# Protective function and durability of mouse lymph node-resident memory CD8+ T cells

Scott M Anthony[1†], Natalija Van Braeckel-Budimir[1†], Steven J Moioffer[1], Stephanie van de Wall[1], Qiang Shan[2,3], Rahul Vijay[2], Ramakrishna Sompallae[1], Stacey M Hartwig[2], Isaac J Jensen[1,2,4], Steven M Varga[1,2,4], Noah S Butler[2,4], Hai-Hui Xue[2,3,4], Vladimir P Badovinac[1,2,4*], John T Harty[1,4*]

[1]Department of Pathology, The University of Iowa, Iowa City, United States; [2]Department of Microbiology and Immunology, The University of Iowa, Iowa City, United States; [3]Center for Discovery and Innovation, Hackensack Meridian Health, Nutley, United States; [4]Interdisciplinary Graduate Program in Immunology, The University of Iowa, Iowa City, United States

*For correspondence:
vladimir-badovinac@uiowa.edu
(VPB);
john-harty@uiowa.edu (JTH)

†These authors contributed
equally to this work

Competing interests: The
authors declare that no
competing interests exist.

Reviewing editor: Gabrielle T
Belz, The University of
Queensland, Australia

**Abstract** Protective lung tissue-resident memory CD8+ T cells (Trm) form after influenza A virus (IAV) infection. We show that IAV infection of mice generates CD69+CD103+ and other memory CD8+ T cell populations in lung-draining mediastinal lymph nodes (mLNs) from circulating naive or memory CD8+ T cells. Repeated antigen exposure, mimicking seasonal IAV infections, generates quaternary memory (4M) CD8+ T cells that protect mLN from viral infection better than 1M CD8+ T cells. Better protection by 4M CD8+ T cells associates with enhanced granzyme A/B expression and stable maintenance of mLN CD69+CD103+ 4M CD8+ T cells, vs the steady decline of CD69+CD103+ 1M CD8+ T cells, paralleling the durability of protective CD69+CD103+ 4M vs 1M in the lung after IAV infection. Coordinated upregulation in canonical Trm-associated genes occurs in circulating 4M vs 1M populations without the enrichment of canonical downregulated Trm genes. Thus, repeated antigen exposure arms circulating memory CD8+ T cells with enhanced capacity to form long-lived populations of Trm that enhance control of viral infections of the mLN.

## Introduction

Influenza A virus (IAV) remains a global heath burden despite long-term worldwide efforts into vaccine development (*Paget et al., 2019*). Current gold-standard vaccinations primarily induce strong and durable antibody responses directed toward the HA and NA proteins of IAV (*Padilla-Quirarte et al., 2019*). However, due to viral antigenic shift and drift, the targets of protective antibody responses are under selective immune pressure and exhibit high mutation rates, reducing the effectiveness of seasonal vaccine approaches (*Thyagarajan and Bloom, 2014*; *Visher et al., 2016*).

Another arm of investigation is the induction of a protective T cell response against conserved sequences across IAV subtypes. Recent reports demonstrate that some level of protective immunity can be acquired in humans against distinct heterosubtypic influenza infections (*McMichael et al., 1983*; *Sridhar et al., 2013*), and animal studies show that this protection can be mediated by memory CD8+ T cells targeting conserved antigens such as the IAV nucleoprotein (NP; *Yewdell et al., 1985*; *Slütter et al., 2013*). Recent evidence suggests a local population of tissue-resident memory CD8+ T cells within the lung parenchyma (lung Trm cells) is associated with this acquired protection following natural or live-attenuated influenza infections (*Wu et al., 2014*; *Van Braeckel-Budimir et al., 2018*; *Slütter et al., 2017*). Unfortunately, and in contrast to the long-lived nature of Trm cells in numerous peripheral and mucosal tissues, Trm cell numbers wane within the lung, and their loss

strongly correlates with the loss of protective immunity to subsequent heterosubtypic IAV infections (*Wu et al., 2014*; *Van Braeckel-Budimir et al., 2018*; *Slütter et al., 2017*).

Despite the annual IAV exposures in humans, the majority of current murine experimental systems rely on one or two sequential IAV infections. In part, this is due to the inability to carry out multiple IAV infections of the same mouse due to the generation of cross-reactive antibodies and the dearth of mouse-adapted influenza strains (*Van Braeckel-Budimir et al., 2017*). To address the impact of repetitive IAV infections on lung Trm, we devised a system where influenza-specific naive or circulating splenic memory CD8$^+$ T cells with a defined number of antigen exposures are isolated and passaged to naive hosts that receive a subsequent intranasal IAV infection (*Van Braeckel-Budimir et al., 2018*). This system generates both lung Trm cells and circulating memory CD8$^+$ T cells that have selectively undergone a defined number of antigenic encounters after lung IAV infection and reflects the repetitive annual infectious nature of influenza within the human population. Utilizing this system, we described key enhancements in both the durability and protective capacity of lung Trm cells generated by repeated influenza antigen exposures (*Van Braeckel-Budimir et al., 2018*).

Recent evidence suggests that in addition to non-lymphoid tissues, populations phenotypically resembling Trm cells are also found within the lymph nodes of both humans and mice, and interestingly, these populations maintain residence during parabiosis experiments (*Schenkel et al., 2014a*; *Beura et al., 2018*; *Kumar et al., 2017*). These cells can be generated in tissue dLN after local infection or derive from pre-existing tissue Trm cells (*Beura et al., 2018*; *Park et al., 2018*). Trm cells in non-lymphoid tissues predominantly act as sentinels, able to quickly activate and produce IFNγ upon the recognition of cognate antigen, which simultaneously induces the recruitment of circulating immune cells and activates local innate immunity that further propagates the inflammatory signal (*Schenkel et al., 2014b*; *Schenkel et al., 2013*). Alternatively, selective non-lymphoid Trm populations are described to have direct cytotoxic capabilities (*Cheuk et al., 2017*). However, it remains unknown if LN Trm cells possess similar properties to those described for non-lymphoid Trm populations and whether LN Trm can mediate protective immunity.

Consistent with a recent study, we show that primary IAV infection generates Trm cells within the lung-draining mLN and that these cells mirror the cell surface marker expression and residence of Trm cells within the lung (*Suarez-Ramirez et al., 2019*; *Stolley et al., 2020*). Importantly, we find that repeated IAV encounters impart numerous changes in the transcriptional and functional landscape of LN Trm cell populations, including enhanced ability to defend the lymph node against viral infection. Unexpectedly, we also observe that multiple stimulated circulating memory CD8$^+$ T cell populations are enriched for the expression of genes whose upregulation is associated with non-lymphoid tissue Trm cells. These data suggest that multiple antigen encounters poise circulating memory CD8$^+$ T cell populations for rapid conversion to durable Trm cells in response to tissue infections, a concept with direct relevance toward effective vaccine design.

## Results

### Respiratory but not systemic infections efficiently induce Trm cells in lung dLN

Following the resolution of systemic or local viral infections, Trm cells are generated within secondary lymphoid organs (SLO Trm cells; *Beura et al., 2018*). Populations of unknown origin, yet phenotypically resembling Trm cells, are also found within numerous human SLOs (*Kumar et al., 2017*). For example, a recent study described the generation of TCR-Tg Trm in lung-draining mediastinal lymph nodes (mLNs) after primary IAV infection of mice (*Stolley et al., 2020*). To address the generality of this observation, we infected naive C57BL6 (B6) mice IN with the IAV X31 (H3N2) and analyzed endogenous NP366 MHC class I tetramer$^+$ cells in the mLN and non-draining cervical lymph nodes (cLNs) 90 days later (*Figure 1A*). Trm express CD69 and in some tissues co-express CD103. As previously described after systemic LCMV infection (*Beura et al., 2018*), CD69$^+$ CD103$^-$ NP366 tetramer$^+$ cells were observed in both mLN and cLN (*Figure 1B*). Consistent with the prior report with TCR-Tg T cells, we also observed an additional population of CD69$^+$ CD103$^+$ NP366 tetramer$^+$- cells primarily localized within the mLN (*Figure 1B*). Thus, IAV infection generates CD69$^+$ CD103$^-$ and CD69$^+$ CD103$^+$ mLN T cells from the endogenous repertoire.

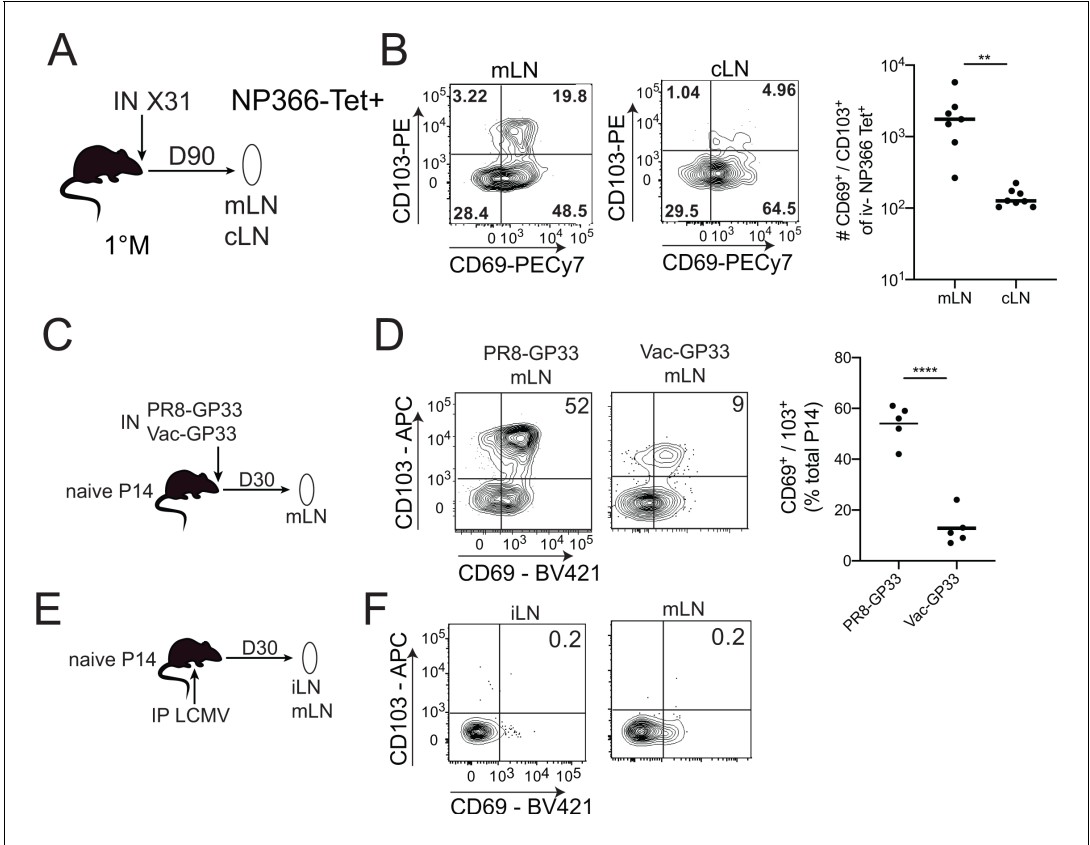

**Figure 1.** CD103+ memory CD8+ T cells are generated in draining lymph nodes (LNs) during localized but not systemic infections. (**A**) C57BL/6 mice were infected IN with X31 (H3N2); mice were sacrificed 90 days post-infection; non-draining cervical lymph nodes (cLN) or lung-draining mediastinal lymph nodes (mLNs) were harvested and analyzed by flow cytometry. (**B**) Representative plots of % of CD69 and CD103 expression (left) in NP366 tetramer+ IV- memory CD8+ T cells from the cLN or mLN and cumulative data (right). n = 3–5 mice/group. Representative of three independent experiments. Bars denote mean values, dots represent independent mice. ****p<0.0001, Students t-test. (**C**) Mice were seeded with 10^4 naive P14 cells and infected IN with either PR8-GP33 (H1N1) or Vac-GP33. 30 days post-infection, draining mLNs were isolated and CD69+ CD103+ P14 Trm populations were evaluated (**D**). Representative plots (left), cumulative data (right). Representative of two independent experiments, n = 5 mice/group. Error bars represent mean ± SD. ****p<0.0001, Students t-test. (**E**) Mice were seeded with 10^4 naive P14 cells and infected IP with LCMV Armstrong. 30 days post-infection LNs (mLN and iLN) were isolated and evaluated for the frequency (**F**) of CD69+/CD103+ P14 s.

The online version of this article includes the following source data for figure 1:

**Source data 1.** Source data for *Figure 1B*.
**Source data 2.** Source data for *Figure 1D*.

To further address the generality and uniformity of these results, Thy 1.2 B6 mice were seeded with naive Thy1.1 P14 TCR-Tg T cells (specific for LCMV GP33-41) and infected IN with either IAV PR8-GP33 or VacV-GP33 (*Figure 1C,D*) or IP with LCMV (*Figure 1E*). Both IN infections generated CD69+ CD103 and CD69+ CD103+ P14 s within the mLN at 30 days post-infection; however, the frequency of CD69+ CD103+ P14 cells was lower in Vac-GP33 infected mice (*Figure 1D*). In contrast, CD69+ CD103+ Trm were not generated in the mLN or iLN after systemic LCMV infection, although we did find the previously described (*Stolley et al., 2020*) CD69+ CD103 P14 in these SLO (*Figure 1F*). Thus, CD69+ CD103+ mLN Trm are not uniformly generated in response to infection, suggesting that the respiratory route of infection may be critical for these populations. Confirmation of this notion will require the evaluation of additional infection and immunization models. These data demonstrated that both endogenous and TCR-transgenic CD69+ CD103+ memory CD8+ T cells were generated specifically in the lung-draining mLN following two intranasal but not a systemic viral infection and that generation of these populations was not limited to IAV infection.

## Protective functions of 1M and 4M LN memory CD8[+] T cells

It is notable that the local protective capacity of LN memory CD8[+] T cells remains undefined, although these cells can be reactivated and migrate to the lung parenchyma after IAV challenge (*Paik and Farber, 2021*). Interrogating this question is technically challenging, because direct rechallenge experiments by the same route of infection result in the pathogen initially encountering protective Trm in the primary infected tissue before reaching the dLN. This issue is further complicated in the IAV model, although virus can be detected in the mLN at 24 hr post-infection of naive hosts (*Flynn et al., 1999*). IAV does not efficiently replicate in this tissue making assessment of virus clearance by memory CD8[+] T cells problematic. An additional issue is that the recurring nature of IAV infections means that humans are likely to experience multiple infections causing their memory T cells to experience repetitive antigen encounters, whereas mouse studies are generally restricted to one or two sequential IAV infections. To address these two issues, we employed a serial adoptive transfer model of P14 circulating naive or memory CD8[+] T cells followed by IN IAV infections (*Van Braeckel-Budimir et al., 2018*) to generate mice with mLN memory P14 cells that had experienced either a single (1M) or four (4M) distinct antigen encounters after lung infection. Additionally, we took advantage of a prior observation that IP infection of mice with LCMV results in initial virus replication in the peritoneal draining mLN (*Olson et al., 2012*). This challenge system allowed us to bypass lung Trm and probe virus control by memory P14 directly in the mLN.

At 100 days post-infection, 1M and 4M mice had similar frequencies of circulating P14 cells in the blood (*Figure 2A*). Total P14 were ~4× higher in the mLN of 4M vs 1M mice (*Figure 2B*). However, an ~50× increase in the number of CD69[+] CD103[+] mLN P14 was observed in 4M vs 1M mice at this time point (*Figure 2C*). IP challenge with LCMV resulted in high mLN virus titers at 3 days post-infection in naive mice, whereas mice with 1M P14 trended toward reduced virus titers, although this was not statistically discernable. In contrast, mice with 4M P14 exhibited robust control of LCMV in the mLN, with most mice exhibiting undetectable virus titers (*Figure 2D*). A detailed assessment of activation status, cytokine profiles, proliferation, and changes in number of mLN memory CD8[+] T cells early after infection will be required to fully understand differences in protection by 1M vs 4M Trm in the mLN. Of note, 4M mice also reduced virus titers in the spleen (*Figure 2E*) suggesting the possibility that control of LCMV in the LN may decrease virus spread within the host. However, it is also possible that the 4M cells in the spleen may have conferred enhanced immunity in this tissue. Further work will be required to resolve these possibilities.

Efficient control of LCMV in the mLN could potentially be mediated by tissue-residing 4M CD8[+] T cells, could involve recruitment of circulating 4M CD8[+] T cells to the infected tissue, or both. To resolve this issue, naive mice and mice with 4M P14 were treated with vehicle or the S1PR agonist FTY720 prior to LCMV infection to prevent recruitment of circulating 4M P14 cells to the mLN. FTY720 treatment discernably but modestly increased virus titers in the mLN of naive mice (*Figure 2F*). In contrast, FTY720 treatment did not influence virus control by 4M P14. Together, these data show that tissue-localized 4M CD8[+] T cells have the capacity to control virus infection of the mLN without the aid of the circulating memory CD8[+] T cell pool.

## Repeated IAV exposure extends the durability of LN Trm cells

Increased numbers of 4M versus 1M CD69[+] CD103[+] memory CD8[+] T cells in the mLN could result from the number of precursor cells adoptively transferred, or reflect differential maintenance of memory populations with increased antigen exposure history. Indeed, 1M lung CD69[+] CD103[+] CD8[+] T cells wane with time after IAV infection resulting in a loss of heterosubtypic immunity (*Wu et al., 2014*; *Slütter et al., 2017*). However, repeated IAV exposures substantially enhanced the durability of 4M lung CD69[+] CD103[+] CD8+ T cells and prolonged heterosubtypic immunity (*Van Braeckel-Budimir et al., 2018*). Therefore, we investigated the durability of 1M and 4M mLN Trm cells. The total number of 4M P14 within the mLN exceeded the number of 1M at early memory time points (*Figure 3A*), but both populations were similar in number from days 150 to 250 post-infection. Within the CD69[+] CD103[+] compartment, 1M mLN cells showed a steady decline in frequency (*Figure 3B*) and total numbers (*Figure 3C*) from days 30 to 250 post-infection, while 4M CD8[+] T cells were largely maintained in frequency and total numbers over the same time frame. This observation was similar to our prior observations with lung Trm cells (*Van Braeckel-Budimir et al., 2018*; *Slütter et al., 2017*), suggesting similar extended survival properties of the lung and mLN 4M Trm

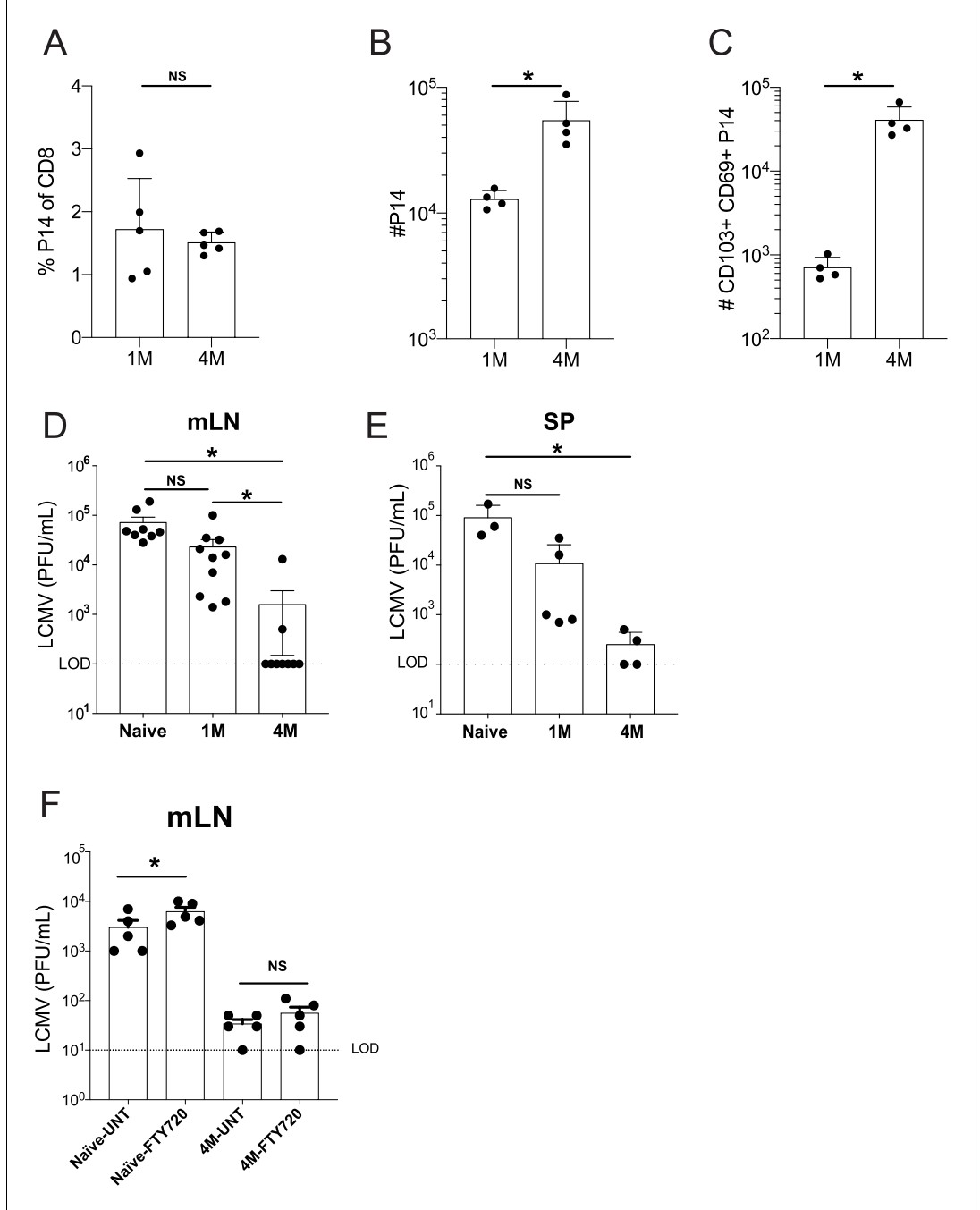

**Figure 2.** Lung-draining LN Trm cells mediate local protective immunity. Mice were seeded with $10^4$ naive or $10^5$ 3M P14 cells and IN infected with PR8-GP33 virus. At 50 days post-infection, frequency of 1M or 4M P14s were measured in the peripheral blood (**A**). At 100 days post-infection, mLNs were harvested and the numbers of total (**B**) and CD103$^+$ CD69$^+$ (**C**) P14s were determined. Representative of three independent experiments, n = 4–5 mice/group. Error bars represent mean ± SD, *p<0.05, **p<0.01, Students t-test in (A–C). LCMV challenge (**D,E**). Naïve, 1M, or 4M mice were infected with LCMV-Armstrong ($2.0 \times 10^5$ PFU/mouse i.p.); 72 hr post LCMV challenge, mLN (**D**) and SP (**E**) were harvested and individually evaluated for LCMV titers by plaque assay. FTY720 treatment impact on LCMV challenge (**F**). Naïve, 1M, or 4M mice were infected with LCMV-Armstrong ($2.0 \times 10^5$ PFU/mouse i.p.) and treated with vehicle or FTY720 daily for 72 hr. 72 hr post LCMV challenge, mLN (**F**) were harvested and individually evaluated for LCMV titers by plaque assay. Dotted line denotes limit of detection. One representative of 2–3 independent experiments is shown, n = 3–5 mice/group. Error bars represent mean ± SEM (**G**) or mean ± SD (**H**). NS = not significant, *p<0.05, one-way ANOVA.

The online version of this article includes the following source data for figure 2:

**Source data 1.** Source data for *Figure 2A*.

**Source data 2.** Source data for *Figure 2B*.

**Source data 3.** Source data for *Figure 2C*.

**Source data 4.** Source data for *Figure 2D*.
**Source data 5.** Source data for *Figure 2E*.
**Source data 6.** Source data for *Figure 2F*.

cell populations derived from circulating 3M precursors. In contrast, total numbers of 1M P14s greatly exceeded those of 4M within the non-draining iLN (*Figure 3—figure supplement 1A*) at all time points. Similar to the endogenous population, CD103$^+$ Trm cells were not observed in the non-draining iLN, with minimal frequencies of CD103$^+$ Trm cells among either 1M or 4M populations (*Figure 3—figure supplement 1B*). Thus, analogous to lung parenchyma Trm cells, IAV-induced CD69$^+$ CD103$^+$ memory cells in the draining mLN wane with time, and repeated antigen exposures substantially extend the durability of this mLN memory CD8$^+$ T cell population. Importantly, the increased durability of CD69$^+$ CD103$^+$ 4M CD8$^+$ T cells at day 100 post-infection likely played an important role in their enhanced ability to control local virus infection.

## Effector functions and localization of mLN memory CD8$^+$ T cells

Protection by memory CD8$^+$ T cells can also depend on their capacity to elaborate antiviral effector mechanisms such as cytolysis and cytokine production (*Martin and Badovinac, 2018*). To address this issue, we initially compared the expression of the cytolytic granule proteins Granzyme (Grz) A and B in 1M and 4M P14 cells. Consistent with their heightened protective capacity, CD103$^+$ and CD103$^-$ P14 4M mLN populations had higher frequencies of GrzB$^+$ and GrzA$^+$ B$^+$ cells than the corresponding 1M P14 mLN populations (*Figure 4A–C*). In contrast, 4M mLN P14 cells exhibited a modest but discernable reduction in the capacity to degranulate and a more pronounced reduction in the capacity to produce IFNγ compared to 1M P14 mLN cells after antigen stimulation (*Figure 4— figure supplement 1A–C*). Additional studies will be required to determine if 4M Trm have heightened killing capacity compared to 1M Trm and if this contributes to the enhanced control of LCMV in the mLN.

Multiple antigen encounters after systemic infections altered many genes in circulating memory CD8$^+$ T cells, including those that control localization (*Wirth et al., 2010*). Therefore, we examined the spatial positioning of 1M and 4M within the LNs. Mice harboring mixed populations of congenically distinct 1M and 4M P14s after IAV infection were subjected to bolus i.v. monoclonal antibody administration (*Mesin et al., 2020*), and we subsequently performed ex vivo two-photon imaging of the intact lymph nodes and reconstructed the tissue via tiling. 1M cells (red dots) were distributed within both the draining mLN (*Figure 5A,B*) and within the non-draining iLN (*Figure 5C,D*). Within the mLN, 4M cells (white dots) were readily abundant and more widely distributed than 1M cells within the mLN (*Figure 5A,B*) but largely absent from the iLN (*Figure 5C,D*). Overall, these data suggest that repeated IAV antigen exposures imparted a broader distribution of 4M CD8$^+$ T cells within the lung-draining mLN. It will be of interest to determine the precise localization (e.g., T cell zones or subcapsular sinus) occupied by CD69$^+$ CD103$^+$ mLN Trm in general and if this distribution is altered for 4M vs 1M Trm as their localization could be important in the control of virus infections entering the mLN (*Reynoso et al., 2019*). Additional studies will be required to determine if enhanced antiviral control by 4M vs 1M P14 mLN cells results from altered mLN distribution.

## CD69$^+$ CD103$^+$ mLN P14 cells are Trm

Trm cells are largely defined by their lack of ability to exit the respective tissue into the circulation (*Klonowski et al., 2004*). Prior studies have suggested that CD69$^+$ CD103$^=$ mLN memory CD8$^+$ T cells are largely but not exclusively Trm (*Beura et al., 2018*). To gain insights into the residential nature of mLN CD69$^+$ CD103$^+$ T cells, we surgically conjoined IAV-experienced mice containing congenically distinct populations of circulating and mLN 1M (Thy 1.1/1.1) and 4M (Thy 1.1/1.2) P14s (*Figure 6A*). After 3 weeks of parabiosis, we examined both the draining (mLN) and non-draining (iLN) for the ability of mLN Trm cells to seed distal tissues and whether multiple antigenic encounters influenced the trafficking of mLN Trm cells. As previously reported (*Van Braeckel-Budimir et al., 2018*), at equilibrium (3 weeks after joining) ~75% of blood and spleen P14 cells in both hosts were derived from the 1M parabiont, while the remaining ~25% were derived from the 4M mouse (data not shown; *Van Braeckel-Budimir et al., 2018*). We also observed P14 cell disequilibrium in the

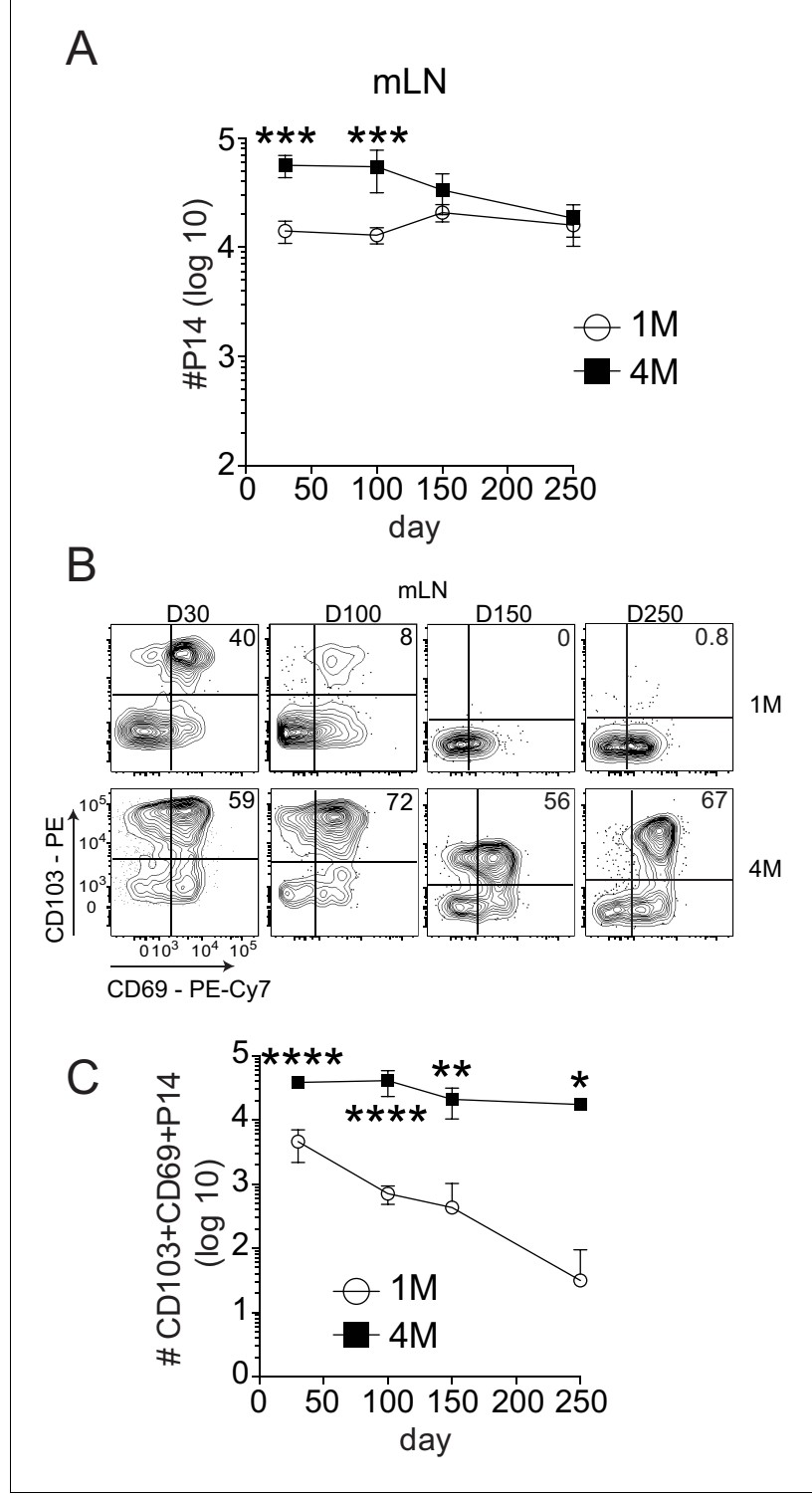

**Figure 3.** Repeated antigen stimulation extends the survival of LN Trms. Mice were seeded with $10^4$ naive or $10^5$ 3M P14 cells and IN infected with PR8-GP33 virus. At indicated time points, mLN (**A**) was harvested and total numbers of 1M (white) and 4M (black) P14 cells were evaluated. Representative of three independent experiments, n = 4 mice/group/time point. Error bars represent mean ± SD. ***p<0.001, ****p<0.0001, two-way ANOVA with Sidak's multiple comparison test. Representative plots (**B**) and cumulative results (**C**) of 1M and 4M CD69+ CD103+ P14 Trm cells in mLN evaluated at indicated time points. Representative of three independent

*Figure 3 continued on next page*

*Figure 3 continued*

experiments, n = 4 mice/group/time point. Error bars represent mean ± SD. *p<0.05, **p<0.01, ****p<0.0001, two-way ANOVA with Sidak's multiple comparison test.

The online version of this article includes the following source data and figure supplement(s) for figure 3:

**Source data 1.** Source data for *Figure 3A*.
**Source data 2.** Source data for *Figure 3C*.
**Figure supplement 1.** CD103⁺ LN Trm cells are not present within non-draining iLN.
**Figure supplement 1—source data 1.** Source data for *Figure 3—figure supplement 1A*.

non-draining iLNs, with >95% derived from the 1M parabiont in both hosts (*Figure 6B*). This is consistent with our prior findings that repetitive antigen encounter results in dramatically delayed acquisition of the LN homing receptor CD62L (*Wirth et al., 2010*; *Nolz and Harty, 2011*). In direct contrast, 4M mLNs contained higher frequencies of total P14s than the 1M mLNs (*Figure 6C*, left). Within the mLN of the 4M parabiont, the 1M P14s were readily abundant within the CD69⁺ CD103⁼ fraction, in agreement with previous reports suggesting that CD69 expression is insufficient to imply residency (*Steinert et al., 2015*). In contrast, P14s co-expressing CD69 and CD103 were largely derived from the respective host populations within the 1M and 4M parabiont mLNs (*Figure 6C*), suggesting these populations were predominantly bone-fide Trm. Therefore, CD69⁺ CD103⁺ cells, whether 1M or 4M, had limited movement from their respective hosts, while the CD69⁺ CD103⁻ 1M population efficiently seeded the mLN of the 4M parabiont, suggesting that CD103 expression, but not CD69 expression, was sufficient for maintaining residence within the mLN after IAV infection.

## Repeated antigen stimulation alters the phenotype of LN Trm cells

The route of infection or the number of antigen encounters can have profound impacts on the phenotype and protective capacity of the resultant population of memory CD8⁺ T cells (*Van Braeckel-Budimir et al., 2018*; *Beura et al., 2018*; *Park et al., 2018*; *Wirth et al., 2010*; *Nolz and Harty, 2011*). Thus, we further defined the phenotype of the mLN T cell populations. The majority of 1M and 4M P14s within non-draining iLN were CD62L⁺ (*Figure 4—figure supplement 2A*). Similarly, 1M within the draining mLN were also mainly CD62L⁺. Consistent with our previous reports (*Van Braeckel-Budimir et al., 2018*; *Wirth et al., 2010*), most 4M within the mLN were CD62L⁻ (*Figure 4—figure supplement 2A*). As previous studies from our group determined that numerous differences between memory CD8⁺ T cells populations in the lung were correlated with the expression or absence of CD103 (*Van Braeckel-Budimir et al., 2018*; *Slütter et al., 2017*), we differentiated the expression of additional markers relative to CD103. Overall, the cell surface expression profile of 1M and 4M CD103⁺ Trm cells was similar to lung Trm cells (*Van Braeckel-Budimir et al., 2018*; *Slütter et al., 2017*), with reduced expression of the shared IL-2/15RB chain CD122 (*Figure 4—figure supplement 2B*) and the transcription factor Eomes (*Figure 4—figure supplement 2C*) and enhanced expression of the alpha subunit of VLA-1 CD49a (*Figure 4—figure supplement 2D*) relative to their CD103⁻ counterparts. The chemokine receptors CXCR3 and CXCR6, which are integral for the migration of CD8⁺ T cells to the lung parenchyma (*Slütter et al., 2013*) and airways (*Hayward et al., 2020*), respectively, were largely similar between these LN populations (*Figure 4—figure supplement 2E,F*). Overall, mLN CD103⁺ Trm cells exhibit a profile of cell surface and transcription factor protein expression similar to bona-fide Trm cells, with both the CD103⁺ and CD103⁻ populations of 4M exhibiting a phenotype similar to 1M CD103⁺ Trm cells.

## Circulating 4 M cells express a core upregulated Trm cell gene signature

In addition to enhancing the durability of mLN Trm cells, repeated IAV exposures also imparted several Trm cell characteristics, such as reduced expression of Eomes and enhanced expression of CD49a to the circulating 4M population (data not shown). As these circulating (Tcirc) 4M or mLN Trm cells may have many additional unique characteristics that promote cellular survival, localization, or their relative abilities to contribute to protective immunity, we sort-purified the respective populations of 1M and 4M splenic Tcirc (*Figure 7—figure supplement 1A*) and mLN Trm cells (*Figure 7—figure supplement 1B*) and analyzed gene expression by bulk RNA-seq (*Table 1*). We initially

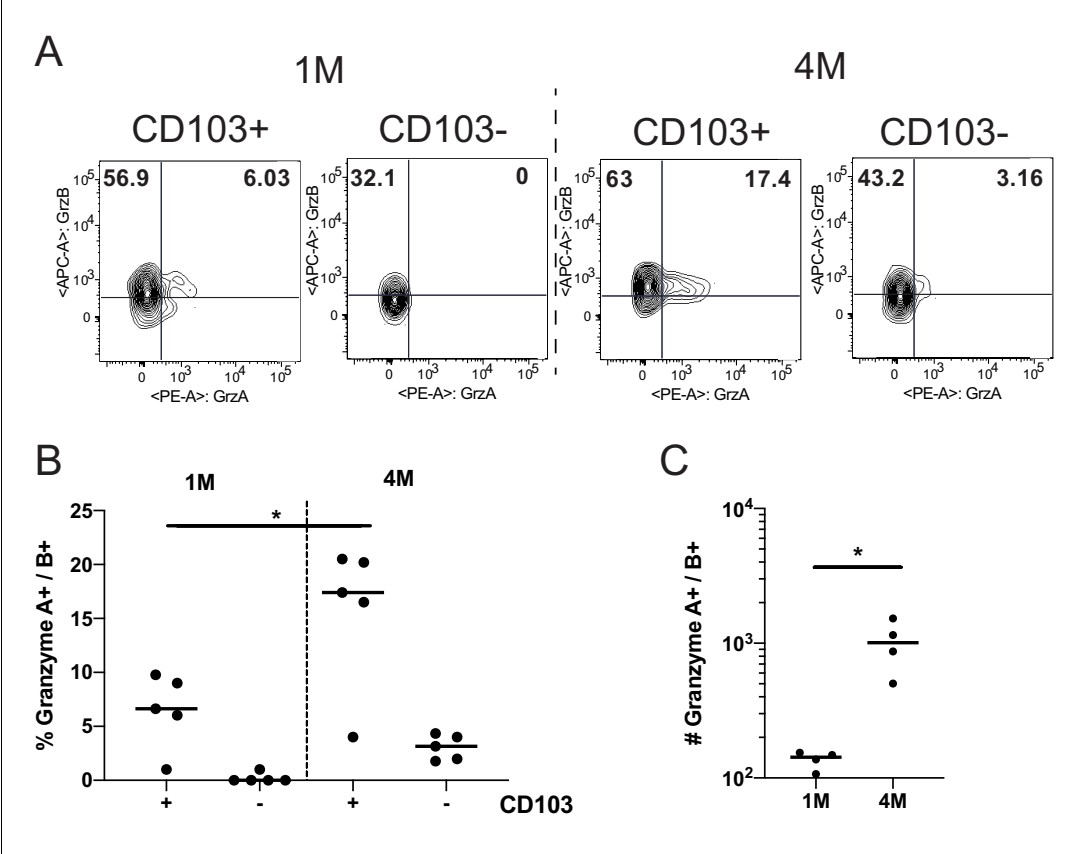

**Figure 4.** Repeated antigen stimulation increases granzyme production of LN Trms. Mice were seeded with $10^4$ naive or $10^5$ 3M P14 cells and IN infected with PR8-GP33 virus. At >60 days post-infection, mLNs were harvested, stimulated with cognate (GP33) peptide in the presence of BFA for 5 hr, and ICS was performed to evaluate the frequency of GrzA$^+$ and GrzB$^+$ fractions of IV$^-$ CD103$^-$ or CD103$^+$ 1M or 4M by flow cytometry. Representative flow plots (A), cumulative frequencies (B), and total numbers per mLN (C) are shown. Representative of two independent experiments, n = 4–5 mice/group. Error bars represent mean ± SD. *p<0.05, one-way ANOVA in (B), *p<0.05, Students t-test in (C).

The online version of this article includes the following source data and figure supplement(s) for figure 4:

**Source data 1.** Source data for *Figure 4B*.

**Source data 2.** Source data for *Figure 4C*.

**Figure supplement 1.** Repeated influenza stimulation reduces cytokine production, but does not affect degranulation capacity of LN Trm cells.

**Figure supplement 1—source data 1.** Source data for *Figure 4—figure supplement 1C*.

**Figure supplement 2.** Repeated antigen stimulation alters the phenotype of LN Trm cells.

**Figure supplement 2—source data 1.** Source data for *Figure 4—figure supplement 2A*.

**Figure supplement 2—source data 2.** Source data for *Figure 4—figure supplement 2B*.

**Figure supplement 2—source data 3.** Source data for *Figure 4—figure supplement 2C*.

**Figure supplement 2—source data 4.** Source data for *Figure 4—figure supplement 2D*.

**Figure supplement 2—source data 5.** Source data for *Figure 4—figure supplement 2E*.

**Figure supplement 2—source data 6.** Source data for *Figure 4—figure supplement 2F*.

focused on the most differentially expressed genes (approximately 1300) between the 1M and 4M LN Trm cells and compared their expression across all four groups. Unbiased hierarchical clustering separated these transcriptional networks into six distinct gene signatures (*Figure 7A*, gene signatures I–VI, inset right). A set of upregulated mRNAs common to both Tcirc and mLN Trm 4M populations (I) contained numerous genes previously identified as being enriched in *Listeria monocyogenes*-induced splenic 4M Tcirc (*Wirth et al., 2010*), including the TNF-R ligands OX40L (*Tnfsf4*), CD30L (*Tnfsf8*), and 41BBL (*Tnfsf9*) and several killer cell lectin-like receptors (KLRs) largely implicated in the stimulation, survival, and NK-like killer functions of expressing lymphocytes (*Martinet and Smyth, 2015*). An upregulated gene set within both Tcirc and mLN Trm 1M

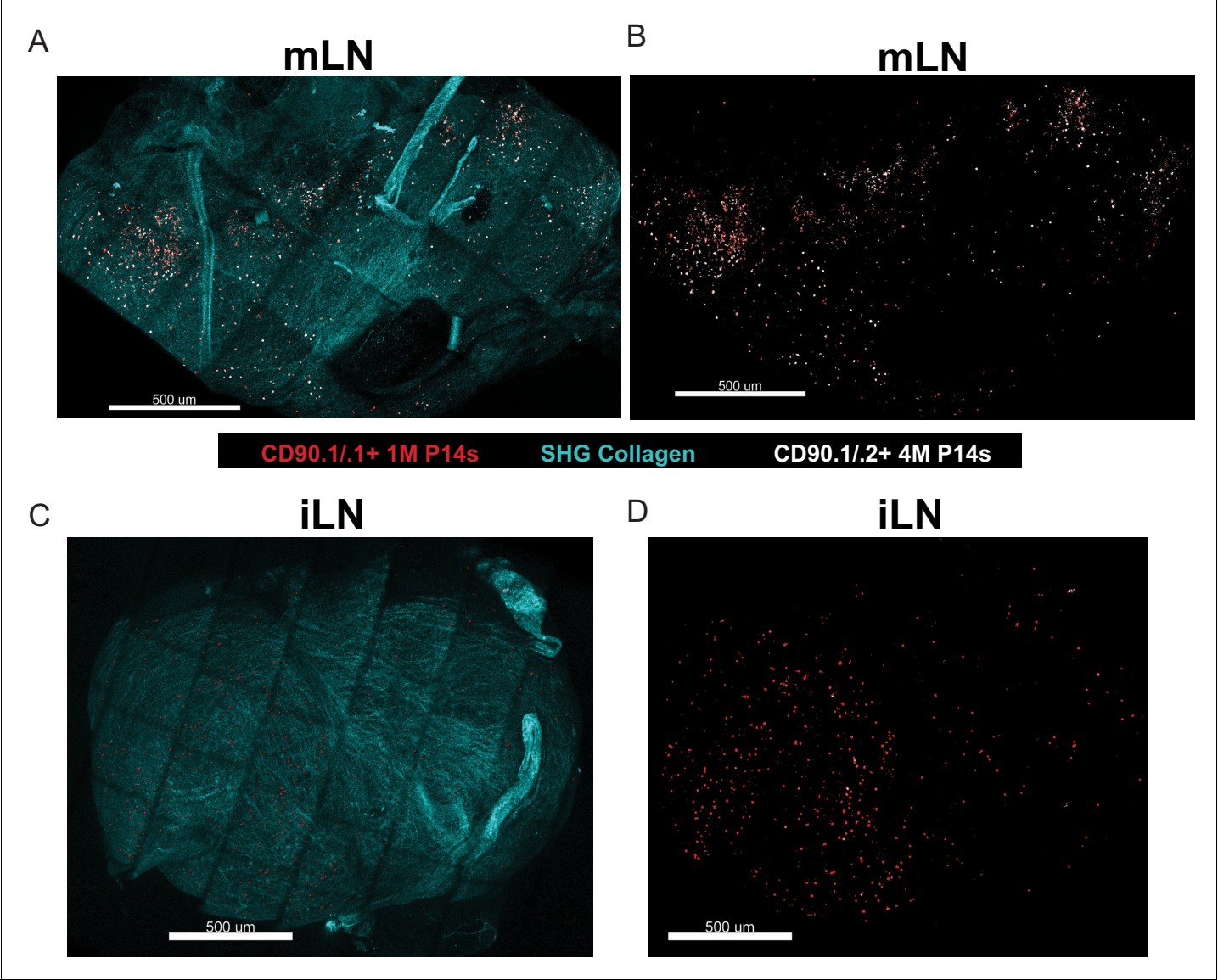

**Figure 5.** Repeated antigen stimulation alters localization of LN Trm cells. (**A–D**) Mice containing mixed congenically distinct populations of 1M (CD90.1/.1, seeded with $10^4$ naive P14) and 4M (CD90.1/.2 seeded with $10^5$ 3M P14) P14s were injected with bolus IV administration of CD90.1-PE (red) and CD90.2-APC (white). Approximately 5 hr post-injection, organs were isolated and two-photon microscopy was performed on whole mLN (**A–B**) or iLN (**C–D**) explants ex vivo. The LN surrounding collagen capsule (pseudocolored blue) was captured with secondary harmonic generation (SHG). Representative of two independent experiments, n = 3–4 mice/group.

The online version of this article includes the following source data for figure 5:

**Source data 1.** Source data for *Figure 5A–D*.

populations (V) contained a mixture of attenuating transcription factors (*Btla, Id3*), the proapoptotic Bcl2 family member Bim (*Bcl2l11*), and the proliferative cytokine Il2. Two signatures were enriched within 1M LN Trm cells. 1M LN Trm cells exhibited enhanced expression of several granzymes (*Grzb, Grzc*) and another proapoptotic Bcl2 family member *Bid* (gene set up IV), while a small number of downregulated 1M Trm cell genes identified (gene set down III) included several factors (*Eomes* and Klrg1) whose downregulation is associated with Trm cells generation (*Mackay et al., 2013*). 4M LN Trm also exhibited two enriched gene signatures. Genes upregulated within 4M LN Trm cells included the antiapoptotic TF *Bcl2*, the effector *Grzf*, and a cytokine (*Csf1*) and chemokine (*Ccl2*) largely responsible for the survival and recruitment of monocytes (*Guilliams et al., 2018*; gene set up II). Of note, despite maintaining residence within the mLN, the 4M LN Trm cells exhibited

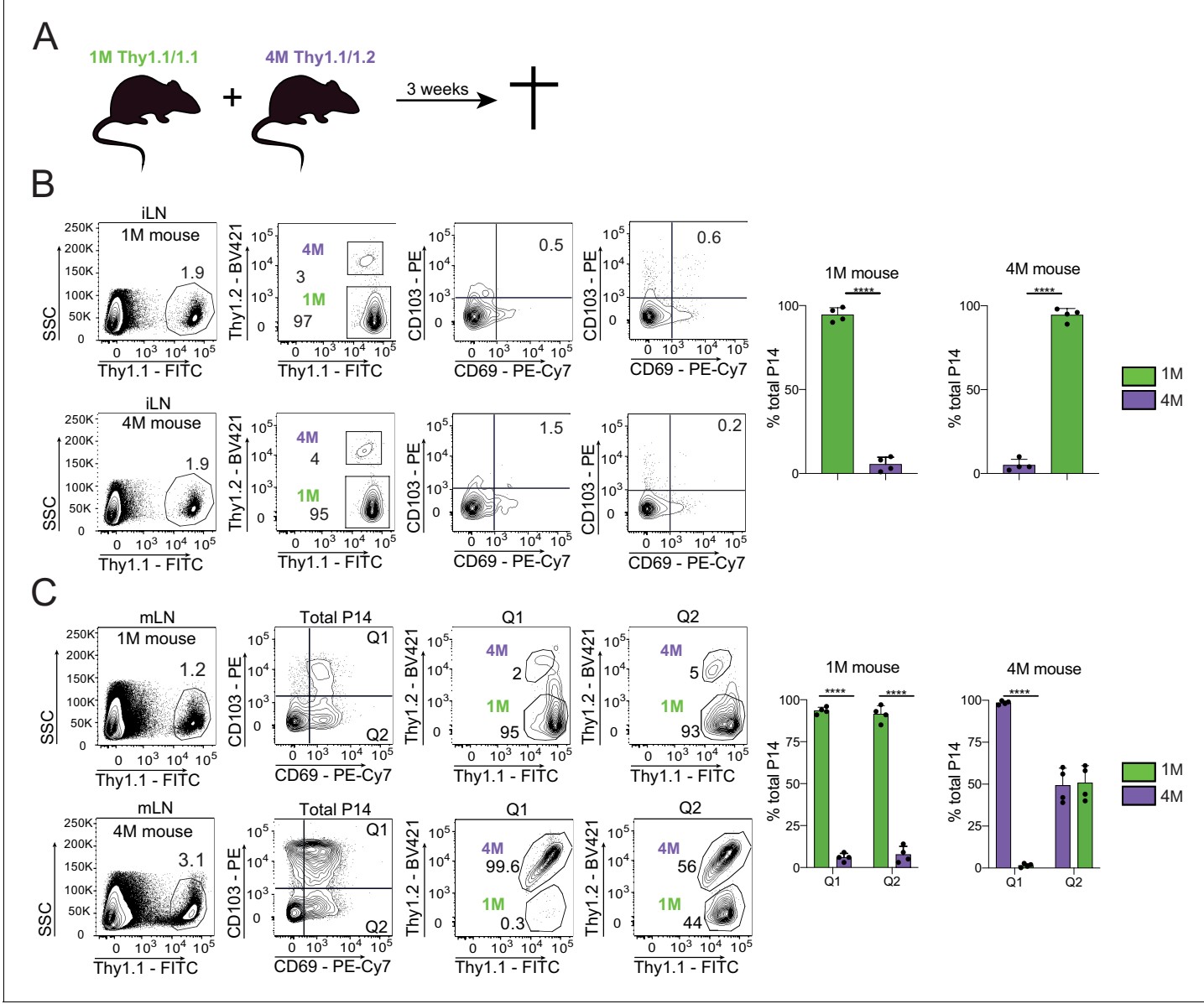

**Figure 6.** Residential nature of 1M and 4M LN Trm cells primed by influenza infection. (**A**) 90 days after IN PR8-GP33 infection, mice bearing Thy1.1/1.1 1M P14 (green; 1M mice seeded with 10⁴ naive P14) cells were joined by parabiotic surgery with mice bearing Thy1.1/1.2 4M P14 (purple, 4M mice seeded with 10⁵ 3M P14). Three weeks later parabionts were analyzed. (**B**) Abundance of 1M (green) and 4M (purple) P14 cells in iLNs of 1M (top row) and 4M (bottom row) parabiotic mice. Representative plots (left), cumulative data (right). Representative of two independent experiments, n = 4 parabionts/experiment. Error bars represent mean ± SD. ****p<0.0001, t-test. (**C**) Abundance and distribution of 1M (green) and 4M (purple) Trm P14 cells expressed as a % of the total Trm population (CD69⁺/CD103⁺) in mLN of 1M (top row) and 4M (bottom row) parabiotic mice. Representative plots (left), cumulative data (right). Representative of two independent experiments, n = 4 parabionts/experiment. Error bars represent mean ± SD. Two-way ANOVA with Sidak's multiple comparison test. Q1(1M) vs Q1(4M) ****p<0.0001; Q2(1M) vs Q2(4M) ****p<0.0001; Q1(1M) vs Q1(4M) ****p<0.0001. The online version of this article includes the following source data for figure 6:

**Source data 1.** Source data for *Figure 6B*.
**Source data 2.** Source data for *Figure 6C*.

reduced expression of numerous genes involved in the trafficking to or retention within lymphoid tissues (gene set down VI), including *Sell* (encoding for CD62L), *Klf2*, *Klf3*, and *Ccr7*.

Although not entirely surprising, many Trm cell-associated genes were not differentially expressed between 1M LN Trm cells and 4M LN Trm cells; however, upon further investigation, we

**Table 1.** Samples for RNAseq, cell number, and RNA quality.

| RNAseq sample | Information | | | | |
|---|---|---|---|---|---|
| Sample info | (pg/μL) RNA concentration | Sample volume (μl) | Total ng, RNA per tube | (RIN) RNA Integrity Number as determined by Agilent | RNA ratio |
| 1M Sp DN #1 | 618 | 17 | 10.506 | N/A | 2 |
| 1M SP DN #2 | 678 | 17 | 11.526 | N/A | 2 |
| 1M SP DN #3 | 1018 | 17 | 17.306 | N/A | 2 |
| 4M Sp DN #1 | 653 | 17 | 11.101 | 9.3 | 2.2 |
| 4M SP DN #2 | 692 | 17 | 11.764 | 9.6 | 2.3 |
| 4M SP DN #3 | 1028 | 17 | 17.476 | 9.4 | 2.1 |
| 1M mLN Trms | 78 | 17 | 1.326 | 9.2 | 1.8 |
| 1M mLN Trms | 139 | 23 | 3.197 | 7.4 | 1.8 |
| 4M mLN Trms | 1200 | 17 | 20.4 | 9.1 | 1.8 |
| 4M mLN Trms | 818 | 17 | 13.906 | 8.7 | 1.8 |

unexpectedly identified upregulation of multiple Trm cell-associated genes within the 4M Tcirc populations. This prompted us to perform additional analyses on the approximately 4000 differentially expressed genes between the 1M and 4M splenic Tcirc. A volcano plot (*Figure 7B*) demonstrated that nearly 90% of these differentially expressed genes were upregulated within the 4M compared to 1M Tcirc population. To gain insights into the prevalence of Trm cell-associated genes within the 4M Tcirc, we compiled a list of genes preferentially upregulated (Up) or downregulated (Down) within various non-lymphoid tissue CD8[+] Trm cell populations described in the literature (*Urban et al., 2020*; *Table 2*) and utilized these gene sets to perform Gene Set Enrichment Analysis (GSEA) on the RNAseq data from 1M and 4M Tcirc populations. Of note, we observed a large enrichment of core upregulated Trm cell-associated genes in 4M Tcirc compared to 1M Tcirc (*Figure 7C*, top), but no appreciable enrichment of core downregulated Trm cell genes (*Figure 7C*, bottom). These data show that repeatedly stimulated 4M Tcirc have a strong enhancement of genes, including *Cd101, Zfp683* (Hobit), *Bhlhe40, Fabp5,* and *Prdm1* (Blimp1), whose upregulation is associated with Trm cell development (heatmap, *Figure 7D*). Overall, these data indicate that 4M Tcirc express a gene signature where many Trm cell core genes are upregulated despite their capacity to circulate within the host. Thus, repetitive antigen encounters appear to poise Tcirc for adoption of a highly sustained tissue-resident lifestyle after tissue antigen re-encounter.

## Discussion

Although IAV infection remains a global threat, many aspects of the basic biology of protective memory CD8[+] T cell responses remain unknown, especially in relation to how the seasonal nature of repetitive IAV infections influences the generation and protective capacity of memory CD8[+] T cells. With continued interrogation into the critical components of protective memory responses, it is clear that the function and distribution of Trm cells is more diverse than previously appreciated. Related populations of Trm cells can clearly reside in both peripheral and lymphoid tissues. Our data corroborate these findings, while also illuminating that multiple antigen encounters enhance the baseline capacity of Tcircm to differentiate into a sustainable population of Trm cells in response to subsequent tissue infection. The enhanced upregulation of a core set of Trm cells-associated genes within this circulating population of 4M likely explains the enhanced durability of Trm cell progeny within the mLN in addition to our previous observation of enhanced durability of multiple stimulated Trm cells within the lung (*Van Braeckel-Budimir et al., 2018*). Although not directly tested here, the profound changes in core Trm gene expression likely reflect strong imprinting of epigenetic regulation in memory CD8[+] T cells by repeated antigen encounters. Given their real-world importance in diseases, such as influenza, malaria, and CMV, that cause repeated infections, it will be of interest to use our model system to dissect the mechanisms underlying epigenetic control of repeatedly stimulated memory CD8[+] T cell populations. Whether repeated antigen encounter shapes the ensuing

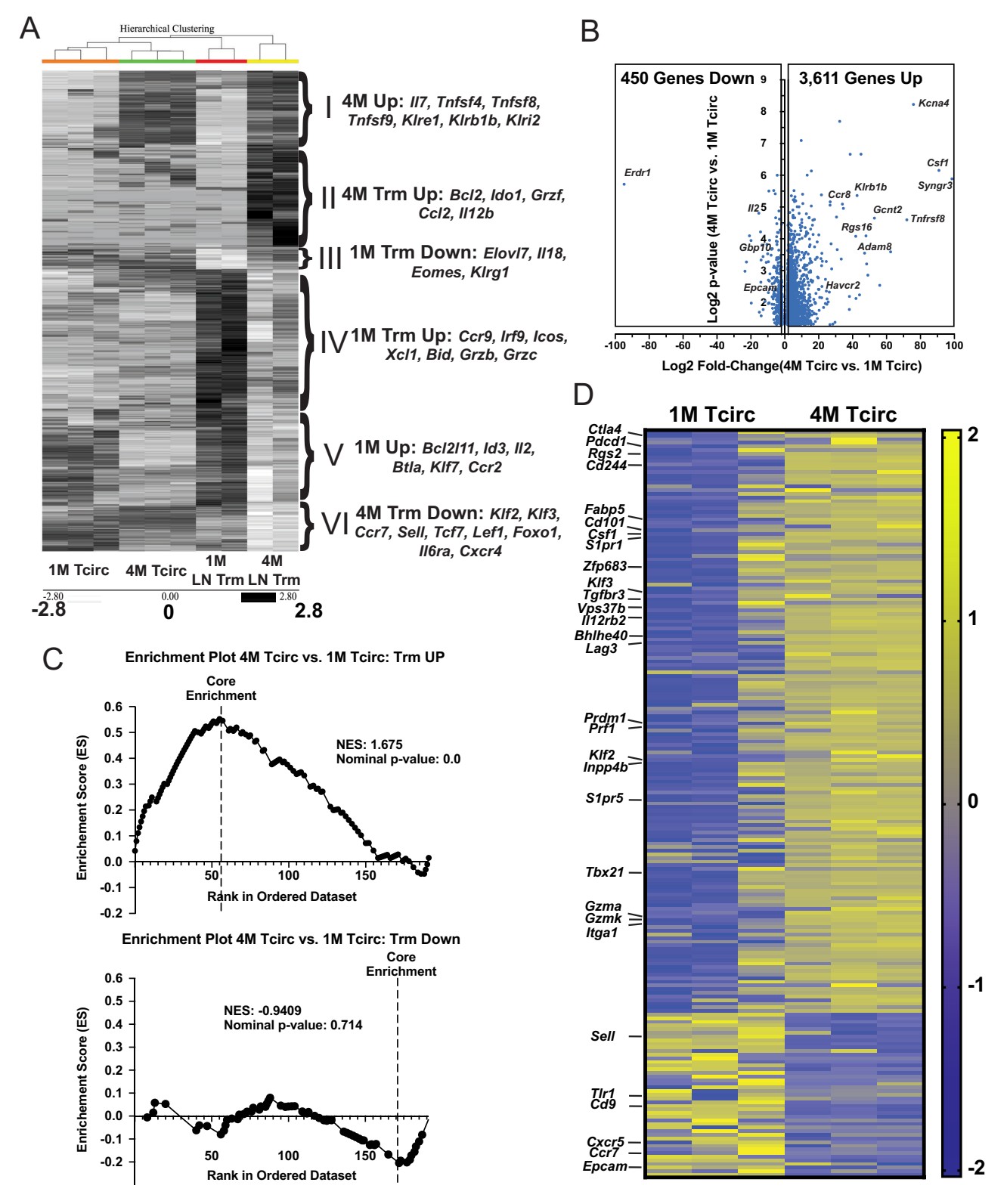

**Figure 7.** Splenic 4M cells express a core Trm signature. Mice were seeded with $10^4$ naive P14 or $10^5$ 3M P14 cells and IN infected with PR8-GP33 virus. At 22–30 days post-infection, IV exclusion was performed and negatively enriched pooled groups of spleens (3–5 spleens/sample, n = 3) or mLNs (15–25 mLNs/sample, n = 2) were stained for CD8α, CD90.1, CD69, and CD103. Bulk RNAseq was performed on RNA from sort-purified spleen samples (20k IV⁻, CD69⁻/CD103⁻ cells/sample) or mLN Trm samples (2–5k IV⁻ CD69⁺/CD103⁺ cells/sample). (**A**) Heatmaps of 1300 most differentially expressed

*Figure 7 continued on next page*

*Figure 7 continued*

genes (log2FC > 1.5, p<0.05) between 1M and 4M LN Trms are plotted from the four respective groups of samples. The six core signature sets of genes offset to the right were derived from unbiased hierarchical clustering. (**B**) Volcano plot of 4061 differentially expressed genes between 1M and 4M splenic memory P14 cells (log2FC > 1.5, p<0.05). (**C**) GSEA of core Trm genes defined in *Table 2* from splenic 1M and 4M populations separated into respective upregulated (top) and downregulated (bottom) gene sets in regard to annotated expression in Trms. (**D**) Heatmap of a core set of selected Trm genes (as in C) within 1M and 4M splenic populations.

The online version of this article includes the following figure supplement(s) for figure 7:

**Figure supplement 1.** Gating strategy of FACS for RNAseq.

memory CD8+ T cell populations by selecting specific precursor populations for further enrichment or by imprinting epigenetic changes on the entire population also remains to determined.

Due to the seasonal and repetitive nature of IAV infections, we speculate that repetitive antigen exposure also occurs within populations of human memory CD8$^+$ T cells specific for conserved influenza-specific antigens. This scenario complicates analyses of human T cell responses and may also influence the response of individuals to vaccines targeting memory CD8$^+$ T cells. Utilizing our system of adoptive transfer and multiple antigenic encounters, the timing, phenotype, and localized nature of LN Trm cells within the draining mLN suggests that their generation is intricately linked with the lung Trm cell population. The overall kinetics of mLN Trm cells were quite similar to the lung Trm cells, which we previously described to wane within the 1M populations and remain largely stable within the 4M populations (*Van Braeckel-Budimir et al., 2018*). As the lung is an intricate tissue involved in gas exchange, it is not unreasonable to speculate that an upper threshold for the number of parenchyma-residing memory cells exists and these additional 4M Trm cells may seed the mLN. A recent study indicates 1M mLN Trm cells may be generated within the lung and drain into the mLN over time via afferent lymphatics (*Stolley et al., 2020*). In part, this process may contribute to the waning of 1M lung Trm and loss of heterosubtypic immunity. Whether 4M lung Trm drain to the mLN remains to be determined. However, the relative stability of lung and mLN 4M Trm compared to 1M Trm may suggest that repeated antigen encounters have the potential to alter the survival and dynamics of Trm generated after IAV infection.

Trm cells are largely characterized within peripheral tissues for their strong antagonism to cues to enter the circulatory and lymphatic systems, yet several studies have indicated that a population of resident memory phenotype CD8$^+$ T cells is present within multiple lymphoid tissues within both mouse models and humans. An underlying question persists why, in addition to circulating Tcm and bona-fide Trm cells, would a host require a resident population of memory T cells within secondary lymphoid tissues? Although an upregulated Trm cell signature was observed within the 4M Tcirc, the reduced mRNA expression encoding lymphoid homing markers (*Ccr7, Sell, Klf2,* and *Klf3*) was only found within the 4M Trm cells, which paradoxically maintain residence within this lymphoid tissue. A study denoting the localization of Trm phenotype cells within the spleen discussed the possibility that they may be capable of providing protection within the SLOs (*Schenkel et al., 2014a*). To test this idea, we employed a model of viral challenge with LCMV i.p., where the initial systemic infection occurs in the same mLN that drain the lungs (*Olson et al., 2012*). In direct experimental support of this notion, we found that 4M LN Trm cells are capable of local protective immunity against a subsequent infection within the mLN in the absence of additional T cell recruitment. Better protective capacity was associated with enhanced maintenance of CD69$^+$ CD103$^+$ 4M compared to 1M CD8$^+$ T cells in the mLN and enhanced local numbers of Granzyme A + B + 4M Trm cells even in the absence of cognate antigen stimulation, suggesting that multiple antigenic encounters poise the memory compartment for rapid production of Granzymes. Populations of circulating CD8$^+$ T cells that have experienced multiple antigen stimulations are associated with enhanced protective capacity against additional challenges with viral infections or tumors (*Jabbari and Harty, 2006*; *Danahy et al., 2020*), and these have been associated with increased granzyme expression (*Danahy et al., 2020*). However, cytotoxic killing capacity is not mutually exclusive from additional potential functions, especially as Trm cells have recently been shown to be capable of giving rise to additional phenotypic lineages of secondary effectors (*Fonseca et al., 2020*). Given that IAV does not efficiently replicate in the mLN, it is fair to ask about the immunological purpose of retaining IAV-specific Trm in this tissue. However, some viruses and bacteria that are acquired by the

**Table 2.** RNAseq heatmaps and GSEA full data.

RNAseq expression related to *Figure 6*.

| Gene | Sp1M-1 | Sp1M-2 | Sp1M-3 | Sp4M-1 | Sp4M-2 | Sp4M-3 |
|---|---|---|---|---|---|---|
| Neurl3 | −1.448372 | −0.744367 | 1.413726 | 0.029106 | 0.558649 | 0.191258 |
| Ctla4 | −1.472581 | −0.893646 | −0.085631 | 0.764674 | 0.93726 | 0.749924 |
| Fn1 | −0.4761 | −0.784447 | 0.313751 | −0.4761 | 1.898997 | −0.4761 |
| Arl4c | −0.219927 | −0.637316 | −0.954088 | −0.368204 | 1.839574 | 0.33996 |
| Pdcd1 | −0.974857 | −1.521079 | 0.271857 | 0.647515 | 0.79075 | 0.785815 |
| Bcl2 | −1.381925 | −0.653217 | 1.527735 | −0.080661 | 0.019993 | 0.568075 |
| Rgs2 | −1.527485 | −0.983776 | 0.344262 | 0.818042 | 0.6478 | 0.701157 |
| Rgs16 | −1.15537 | −1.037752 | −0.469227 | 0.887162 | 0.836029 | 0.939158 |
| Fcgr2b | −0.959014 | −0.670621 | −1.068859 | 0.853631 | 0.752644 | 1.092219 |
| Cd244 | −0.4632 | −0.776816 | −1.37756 | 0.884462 | 0.876998 | 0.856116 |
| Slamf7 | −1.139613 | −1.165538 | −0.294917 | 0.903817 | 0.860608 | 0.835643 |
| Atf3 | −0.779708 | −0.779708 | −0.779708 | 0.517244 | 0.120688 | 1.701193 |
| Il2ra | −1.322889 | −1.127548 | 0.021861 | 0.733615 | 0.889219 | 0.805742 |
| Zeb2 | −1.316736 | −1.060834 | −0.124986 | 0.843957 | 0.692407 | 0.966192 |
| Nr4a2 | −1.166233 | −1.064171 | −0.391073 | 0.70188 | 0.861206 | 1.058391 |
| Itga6 | 0.103598 | −1.354866 | 1.208944 | −1.045011 | 0.477086 | 0.610248 |
| Chn1 | −0.408248 | −0.408248 | −0.408248 | 2.041241 | −0.408248 | −0.408248 |
| Itga4 | −1.533817 | −0.935003 | 0.956091 | 0.590266 | 0.254859 | 0.667605 |
| Cd44 | −0.687106 | −0.641904 | −1.152441 | 0.138632 | 1.061634 | 1.281186 |
| Sema6d | −1.095775 | −1.095775 | −0.277219 | 1.304133 | 0.315154 | 0.849482 |
| Dusp2 | −1.284804 | −1.248205 | 0.363333 | 0.516972 | 0.711724 | 0.94098 |
| Bcl2l1 | −1.599476 | −0.920734 | 0.650549 | 0.563217 | 0.660348 | 0.646096 |
| Fgf13 | −1.184461 | −1.352896 | 0.33382 | 0.580607 | 0.836326 | 0.786603 |
| Cxcr3 | −1.558265 | −0.934718 | 0.89392 | 0.368323 | 0.670138 | 0.560601 |
| Fabp5 | −0.937223 | −0.580515 | −1.161553 | 0.966239 | 0.987902 | 0.72515 |
| Tnfsf10 | −0.757581 | −1.3349 | −0.529128 | 0.782834 | 0.745086 | 1.093689 |
| Rorc | −0.833666 | −0.833666 | 0.249621 | −0.833666 | 0.726727 | 1.52465 |
| Cd101 | −1.953973 | −0.122477 | 0.328694 | 0.424373 | 0.621766 | 0.701617 |
| Csf1 | −1.089549 | −0.89601 | −0.735068 | 0.905082 | 0.872774 | 0.94277 |
| S1pr1 | −1.414071 | −1.10549 | 0.261792 | 0.695359 | 0.834468 | 0.727942 |
| Usp33 | −1.470917 | −1.045478 | 0.725238 | 0.869896 | 0.325855 | 0.595405 |
| Bach2 | −1.176816 | −0.99054 | 1.523135 | 0.417138 | 0.352504 | −0.125421 |
| Aqp3 | −1.330326 | −1.216594 | 0.833799 | 0.371489 | 0.762789 | 0.578843 |
| Coro2a | −1.303957 | −1.114023 | −0.062495 | 0.784956 | 0.872198 | 0.823321 |
| Tnfsf8 | −1.655898 | −0.080202 | 1.478025 | 0.013587 | 0.258145 | −0.013656 |
| Dmrta1 | −0.845576 | −1.149819 | 1.698186 | −0.055975 | 0.094192 | 0.258992 |
| Jun | −1.376227 | −1.162485 | 0.313126 | 0.681802 | 0.765077 | 0.778706 |
| Zfp683 | −1.593545 | −0.902161 | 0.462586 | 0.734325 | 0.491177 | 0.807619 |
| Runx3 | −1.150841 | −1.327161 | 0.987223 | 0.185512 | 0.48935 | 0.815917 |
| Tnfrsf1b | −1.135037 | −1.426812 | 0.622179 | 0.498893 | 0.688943 | 0.751834 |
| Abcb1a | −1.375583 | −1.001921 | −0.109472 | 0.691716 | 0.866081 | 0.929179 |
| Cd36 | −1.132023 | −1.132023 | 0.647012 | 1.350724 | 0.414674 | −0.148364 |
| Fosl2 | 0.574051 | −1.743753 | −0.617194 | 0.260588 | 0.853012 | 0.673296 |

*Table 2 continued on next page*

*Table 2 continued*

| Gene | Sp1M-1 | Sp1M-2 | Sp1M-3 | Sp4M-1 | Sp4M-2 | Sp4M-3 |
|------|--------|--------|--------|--------|--------|--------|
| Cd38 | −1.726006 | −0.546715 | 0.169224 | 0.342229 | 0.768355 | 0.992914 |
| Klf3 | −1.580977 | −0.948517 | 0.612343 | 0.634295 | 0.623061 | 0.659795 |
| Cxcl9 | −0.779708 | −0.779708 | 0.517244 | 1.701193 | 0.120688 | −0.779708 |
| Tgfbr3 | −1.496104 | −1.003156 | 0.272229 | 0.611554 | 0.842089 | 0.773388 |
| Dtx1 | −1.323911 | −0.532375 | 1.651323 | −0.271103 | 0.079677 | 0.39639 |
| Vps37b | −1.327024 | −1.248228 | 0.731573 | 0.523888 | 0.673187 | 0.646604 |
| Hsph1 | −0.89509 | −1.269058 | −0.482019 | 0.763431 | 0.921505 | 0.961231 |
| Chn2 | −1.469041 | −1.039823 | 0.366715 | 0.565777 | 0.966551 | 0.609821 |
| Il12rb2 | −1.512436 | −0.863636 | −0.044159 | 0.717179 | 0.841954 | 0.861097 |
| Cd8b1 | −0.897526 | −1.057029 | −0.74355 | 0.650528 | 1.07277 | 0.974808 |
| Cd8a | −1.369955 | −0.899035 | −0.293171 | 0.691408 | 0.958922 | 0.91183 |
| Sema4f | −0.711384 | −1.581909 | −0.130766 | 0.707607 | 0.841139 | 0.875313 |
| Mxd1 | −1.636166 | −0.835304 | 0.820257 | 0.420078 | 0.520451 | 0.710683 |
| Bhlhe40 | −1.377693 | −0.960477 | −0.194366 | 0.794975 | 0.876639 | 0.860922 |
| Klrg1 | −0.649531 | −0.92849 | −1.127976 | 0.880401 | 0.945822 | 0.879774 |
| Lag3 | −1.314079 | −1.250661 | 0.47627 | 0.562176 | 0.736842 | 0.789452 |
| Klre1 | −0.817609 | −0.924946 | −0.991491 | 0.900879 | 0.893107 | 0.940061 |
| Klrd1 | −0.82806 | −1.237129 | −0.605683 | 0.779021 | 0.84455 | 1.0473 |
| Klrc2 | −1.060961 | −0.22419 | −1.157839 | 1.312817 | 0.318738 | 0.811434 |
| Klrc1 | −0.456844 | −0.628525 | −1.48154 | 0.799308 | 0.874063 | 0.893538 |
| Dusp16 | −1.549192 | −0.590124 | −0.304361 | 0.482539 | 0.940286 | 1.020853 |
| Emp1 | −0.820283 | −1.020728 | −0.88602 | 0.790046 | 0.961756 | 0.975229 |
| Fosb | −1.192929 | −1.192929 | −0.104719 | 0.817943 | 0.642677 | 1.029956 |
| Nkg7 | 0.856662 | 0.552759 | −1.93607 | 0.002826 | 0.455585 | 0.068237 |
| Ppp1r15a | −1.743276 | −0.690768 | 0.651869 | 0.583382 | 0.59713 | 0.601662 |
| Swap70 | −1.185293 | −1.273627 | 0.02884 | 0.760466 | 0.836827 | 0.832787 |
| Il4ra | −1.240991 | −1.270277 | 1.021941 | 0.40852 | 0.380698 | 0.70011 |
| Il21r | −1.376141 | −1.176848 | 0.858231 | 0.430021 | 0.630542 | 0.634194 |
| Itgal | −1.41421 | −1.139199 | 0.44835 | 0.575817 | 0.77812 | 0.751121 |
| Itgax | −1.299963 | −1.141223 | −0.005996 | 0.730109 | 0.843686 | 0.873388 |
| Bag3 | −1.261842 | −1.302585 | 0.579948 | 0.471619 | 0.819237 | 0.693622 |
| Adam8 | −1.448605 | −0.760408 | −0.355088 | 0.802297 | 0.908979 | 0.852826 |
| Ifitm2 | −1.931568 | 0.697426 | −0.221955 | 0.354391 | 0.572163 | 0.529542 |
| Ifitm1 | −0.116103 | −0.604934 | −1.639024 | 0.602024 | 0.9939 | 0.764137 |
| Ifitm3 | −0.399411 | −1.471071 | −0.006047 | −0.164996 | 1.552387 | 0.489138 |
| Ifitm10 | −1.619476 | −0.810367 | 0.198443 | 0.630475 | 0.833945 | 0.76698 |
| Ifngr1 | −1.689527 | −0.724236 | 0.542852 | 0.31725 | 0.873566 | 0.680095 |
| Prdm1 | −0.763724 | −0.527892 | −1.266607 | 1.19839 | 0.974068 | 0.385764 |
| Prf1 | −1.450717 | −1.022122 | 0.129833 | 0.732409 | 0.819384 | 0.791213 |
| Gzmm | −0.148129 | −1.768864 | 0.896369 | −0.297885 | 0.868697 | 0.449812 |
| Gadd45b | −1.732358 | −0.68227 | 0.405453 | 0.592053 | 0.623696 | 0.793427 |
| Phlda1 | −1.336075 | −1.197598 | 0.286457 | 0.783069 | 0.814378 | 0.649768 |
| Ifng | −1.270643 | −0.812118 | −0.587444 | 0.872087 | 0.942033 | 0.856084 |
| A430078g2 | −0.768864 | −1.672411 | 0.352995 | 0.762701 | 0.546624 | 0.778955 |

*Table 2 continued on next page*

*Table 2 continued*

| Gene | Sp1M-1 | Sp1M-2 | Sp1M-3 | Sp4M-1 | Sp4M-2 | Sp4M-3 |
|------|--------|--------|--------|--------|--------|--------|
| Rasa3 | −1.298438 | −1.268276 | 0.456223 | 0.612102 | 0.740735 | 0.757654 |
| Dusp4 | −0.013625 | −1.524651 | −0.248373 | −0.013625 | 1.604724 | 0.19555 |
| Lpl | −0.751462 | −0.751462 | −0.751462 | −0.058456 | 0.596117 | 1.716724 |
| Klf2 | −0.890686 | −1.058738 | −0.622839 | 0.304069 | 1.26258 | 1.005613 |
| Inpp4b | −1.411831 | −1.122956 | 0.917949 | 0.548702 | 0.408487 | 0.65965 |
| Dnajb1 | −1.351654 | −1.187033 | 0.50308 | 0.456015 | 0.956211 | 0.623381 |
| Junb | −1.248952 | −1.324103 | 0.75357 | 0.508582 | 0.679045 | 0.631859 |
| Cdh1 | −1.503072 | −0.854578 | 1.239026 | 0.498073 | 0.17862 | 0.441931 |
| Itgb1 | −1.070625 | −1.246026 | −0.245061 | 0.669811 | 0.908323 | 0.98358 |
| Fut11 | −1.566705 | −0.907343 | 0.264437 | 0.640465 | 0.710643 | 0.858503 |
| Gzmc | −1.129587 | −1.129587 | 1.393745 | 0.701502 | 0.081963 | 0.081963 |
| Gzmb | −1.019828 | −0.852547 | −0.860309 | 0.860605 | 0.936848 | 0.935231 |
| Tmem123 | −0.486668 | −0.568492 | −1.383266 | 0.289273 | 1.332877 | 0.816276 |
| Icam1 | −1.395117 | −1.177656 | 0.622649 | 0.592831 | 0.735718 | 0.621575 |
| S1pr5 | −1.401126 | −1.070617 | 0.107256 | 0.68738 | 0.826563 | 0.850543 |
| Gm10080 | −0.187622 | −0.228053 | −1.797739 | 0.584955 | 0.734187 | 0.894272 |
| Smad3 | −1.296473 | −1.254074 | 0.436554 | 0.494756 | 0.817683 | 0.801554 |
| Anxa2 | −1.161681 | −1.045596 | −0.444054 | 0.777762 | 0.935755 | 0.937813 |
| Cx3cr1 | −1.267574 | −1.170718 | −0.023068 | 0.778225 | 0.846035 | 0.837099 |
| Ccr8 | −1.030745 | −1.030745 | −0.649317 | 0.859499 | 0.902093 | 0.949215 |
| Crr9 | −1.163685 | −1.163685 | 0.518173 | 0.059333 | 1.369219 | 0.380645 |
| Cxcr6 | −1.008609 | −0.713182 | −0.998656 | 0.85504 | 0.986542 | 0.878867 |
| Ccr1 | −0.959366 | −0.959366 | −0.044992 | −0.460137 | 1.148955 | 1.274906 |
| Ccr2 | −1.244353 | −0.427864 | −0.835107 | 0.647598 | 0.470274 | 1.389451 |
| Ccr5 | −1.267395 | −1.197781 | 0.084813 | 0.600219 | 0.830419 | 0.949725 |
| Adam19 | −1.693324 | −0.353517 | 1.307494 | 0.407904 | 0.361716 | −0.030273 |
| Havcr2 | −0.267637 | −1.661471 | −0.479217 | 0.747014 | 0.819872 | 0.841438 |
| Itgae | 0.265873 | −0.131851 | −1.890634 | 0.287854 | 1.030981 | 0.437777 |
| Traf4 | −1.26168 | −1.317019 | 0.664003 | 0.656493 | 0.700534 | 0.557669 |
| Ccl5 | 0.717805 | 0.069708 | −1.965756 | 0.155463 | 0.696447 | 0.326333 |
| Ccl9 | −0.096406 | −1.204224 | −1.176934 | 0.685658 | 1.095686 | 0.69622 |
| Ccl3 | 0.321741 | 0.441288 | −2.033821 | 0.347205 | 0.557351 | 0.366236 |
| Ccl4 | −0.900786 | −1.071795 | −0.743744 | 0.770811 | 0.976086 | 0.969428 |
| Wfikkn2 | −1.521007 | −0.464233 | 1.525421 | −0.037938 | 0.151938 | 0.345819 |
| Tbx21 | −1.188999 | −1.37588 | 0.475345 | 0.610345 | 0.759099 | 0.720091 |
| Arl5c | −1.071685 | −1.481244 | 0.622113 | 0.521195 | 0.755916 | 0.653704 |
| Stat3 | −1.33085 | −1.244827 | 0.568421 | 0.575872 | 0.722103 | 0.709281 |
| Icam2 | −1.427743 | −1.074286 | 0.307995 | 0.501199 | 0.965903 | 0.726932 |
| Cmah | −1.619268 | −0.820081 | 1.012438 | 0.481339 | 0.443919 | 0.501653 |
| Fam65b | −1.435265 | −1.087651 | 0.278384 | 0.756549 | 0.73623 | 0.751753 |
| Irf4 | −0.942848 | −1.402093 | −0.166421 | 0.737652 | 0.874145 | 0.899565 |
| Ly86 | −1.176696 | −1.176696 | 0.271478 | 0.702255 | 0.093094 | 1.286564 |
| Nfil3 | −1.352986 | −1.210674 | 0.442696 | 0.664949 | 0.780773 | 0.675242 |
| Cdc14b | −1.660352 | −0.625526 | −0.018991 | 0.682479 | 0.624463 | 0.997927 |

*Table 2 continued on next page*

*Table 2 continued*

| Gene | Sp1M-1 | Sp1M-2 | Sp1M-3 | Sp4M-1 | Sp4M-2 | Sp4M-3 |
|------|--------|--------|--------|--------|--------|--------|
| Naip3 | −0.645497 | −0.645497 | −0.645497 | −0.645497 | 1.290994 | 1.290994 |
| Elovl7 | −0.792302 | −0.420558 | −1.350578 | 1.107228 | 0.934382 | 0.521828 |
| Gzma | −0.686737 | −0.814558 | −1.197182 | 0.848737 | 0.941036 | 0.908703 |
| Gzmk | −1.406303 | −0.870165 | −0.278402 | 0.764809 | 0.91127 | 0.878791 |
| Itga1 | −1.285811 | −1.102208 | −0.113217 | 0.834949 | 0.703502 | 0.962785 |
| Rhob | −0.230141 | −1.869004 | 0.130554 | 0.363198 | 0.71253 | 0.892863 |
| Id2 | −1.046589 | −0.909419 | −0.760447 | 0.760224 | 0.914986 | 1.041244 |
| Adam4 | 0.587047 | −1.985993 | 0.480645 | −0.035246 | 0.587047 | 0.3665 |
| Fos | −0.879435 | −1.290058 | −0.474026 | 0.882188 | 0.81645 | 0.94488 |
| Il7r | −0.170366 | −0.087313 | −1.825959 | 0.423786 | 0.64009 | 1.019763 |
| Ly6c1 | −0.753065 | −1.398044 | −0.214637 | 0.174751 | 1.067836 | 1.123158 |
| Ly6c2 | −1.276681 | −1.209469 | 0.15969 | 0.548136 | 0.896153 | 0.882171 |
| Il2rb | −1.273008 | −1.30208 | 0.705662 | 0.505892 | 0.665374 | 0.698162 |
| Bin2 | −1.441078 | −1.122664 | 0.546309 | 0.600436 | 0.709577 | 0.70742 |
| Nr4a1 | −1.212606 | −1.312252 | 1.027233 | 0.554366 | 0.462063 | 0.481197 |
| Itga5 | −0.995646 | −0.995646 | 1.538635 | −0.396988 | 0.681175 | 0.16847 |
| Litaf | −1.67839 | −0.692126 | 0.153189 | 0.817433 | 0.827332 | 0.572561 |
| Klhl6 | −1.527286 | −0.908352 | 0.137948 | 0.561523 | 0.888302 | 0.847865 |
| Bcl6 | −1.042258 | −1.315998 | −0.140006 | 0.851054 | 1.025231 | 0.621977 |
| Tigit | −0.636695 | −0.492403 | −1.452925 | 0.80108 | 0.972809 | 0.808135 |
| Sidt1 | −1.416356 | −0.917945 | 1.352363 | 0.320931 | 0.308518 | 0.352489 |
| Ccr6 | −0.408248 | −0.408248 | −0.408248 | −0.408248 | 2.041241 | −0.408248 |
| Hspa1a | −1.489338 | −1.05789 | 0.546575 | 0.59207 | 0.600862 | 0.807722 |
| Tnf | −1.932303 | −0.226197 | 0.37222 | 0.508253 | 0.666342 | 0.611686 |
| Tnfsf9 | −0.721049 | −1.122053 | −0.869307 | 0.871738 | 0.99594 | 0.84473 |
| Qpct | −0.972332 | 0.746934 | −0.972332 | −0.773832 | 0.919129 | 1.052433 |
| Slc3a2 | −1.412175 | −1.03627 | 0.068812 | 0.641712 | 0.919915 | 0.818006 |
| Dtx4 | −1.895821 | 0.031087 | 0.985432 | 0.206382 | 0.623526 | 0.049393 |
| Dusp5 | −1.204462 | −1.231643 | −0.009555 | 0.655736 | 0.90792 | 0.882003 |
| Aff3 | 0.334158 | 0.934136 | 1.302955 | −0.695519 | −1.131655 | −0.744075 |
| Icos | 1.471137 | 0.841393 | 0.145304 | −0.792242 | −1.047218 | −0.618376 |
| Ikzf2 | −0.165671 | 1.704978 | 0.658251 | −0.793835 | −0.609887 | −0.793835 |
| Cxcr4 | 0.869115 | 0.919333 | 0.906884 | −0.886976 | −1.182769 | −0.625587 |
| Cd55 | 0.694467 | 0.444026 | 1.397141 | −0.459842 | −1.074231 | −1.001562 |
| Rgs1 | 0.282992 | −0.028889 | 1.320663 | 0.669655 | −1.444527 | −0.799896 |
| Sell | 0.343005 | 0.577399 | 1.57097 | −0.879322 | −0.741027 | −0.871025 |
| Xcl1 | 1.158069 | 0.726765 | 0.823087 | −0.933921 | −0.948108 | −0.825893 |
| Slamf6 | 0.878115 | 0.648026 | 1.158166 | −0.969274 | −0.688009 | −1.027024 |
| Dapl1 | 0.95876 | 0.953171 | 0.822988 | −0.894936 | −0.945047 | −0.894936 |
| Mal | 0.278549 | −1.166388 | 0.89838 | −0.024742 | 1.180589 | −1.166388 |
| Pmepa1 | 1.000943 | 1.530105 | −0.429028 | −0.700673 | −0.700673 | −0.700673 |
| Fabp4 | −0.408248 | 2.041241 | −0.408248 | −0.408248 | −0.408248 | −0.408248 |
| Skil | 1.47811 | 1.07101 | −0.815616 | −0.553424 | −0.577221 | −0.602859 |
| Il2 | 0.843097 | 1.08096 | 0.786459 | −0.913213 | −0.735725 | −1.061578 |

*Table 2 continued on next page*

*Table 2 continued*

| Gene | Sp1M-1 | Sp1M-2 | Sp1M-3 | Sp4M-1 | Sp4M-2 | Sp4M-3 |
|------|--------|--------|--------|--------|--------|--------|
| Foxo1 | 1.398004 | 1.047696 | −0.261312 | −1.167157 | −0.508615 | −0.508615 |
| Il6ra | 0.192477 | 0.123263 | 1.813008 | −0.551343 | −0.551343 | −1.026062 |
| Npr1 | −0.408248 | 2.041241 | −0.408248 | −0.408248 | −0.408248 | −0.408248 |
| Lef1 | −0.577413 | −0.122582 | 2.008562 | −0.513026 | −0.534901 | −0.26064 |
| Lpar3 | −0.600795 | −0.600795 | 1.779753 | −0.600795 | −0.600795 | 0.623426 |
| Ifi44 | −1.522188 | 1.370539 | 0.77727 | −0.414406 | −0.162093 | −0.049122 |
| Nr4a3 | 1.658887 | −1.434777 | 0.298688 | −0.213395 | −0.096008 | −0.213395 |
| Tlr1 | 1.087014 | −1.759701 | 0.693949 | −0.332077 | −0.04672 | 0.357535 |
| Cxcl10 | 0.819849 | −1.553617 | 1.217093 | 0.268993 | −0.179362 | −0.572955 |
| Cd27 | −0.812702 | 0.736012 | 1.686726 | −0.76644 | −0.35287 | −0.490726 |
| Cd9 | 1.139853 | 0.963004 | 0.584072 | −0.83021 | −0.83021 | −1.02651 |
| Cd69 | 0.624643 | 0.93042 | 1.125309 | −0.853084 | −0.712926 | −1.114363 |
| Isg20 | 1.393478 | 0.934687 | 0.197602 | −0.930029 | −0.737091 | −0.858648 |
| Rgs10 | 0.96634 | 1.037646 | 0.595869 | −0.585759 | −0.644411 | −1.369686 |
| Themis | −0.659209 | 1.957135 | −0.786652 | −0.207908 | −0.035484 | −0.267882 |
| Fyn | 1.066584 | 1.346547 | 0.054683 | −0.726098 | −0.837588 | −0.904128 |
| Egr2 | 1.036188 | −1.636837 | 1.073379 | −0.057847 | −0.260015 | −0.154868 |
| Timp3 | −0.408248 | 2.041241 | −0.408248 | −0.408248 | −0.408248 | −0.408248 |
| Dusp6 | −0.205991 | 0.085139 | 1.646718 | −1.49606 | −0.014903 | −0.014903 |
| Crtam | 1.079203 | 1.14537 | 0.408387 | −1.043006 | −0.745838 | −0.844116 |
| Cxcr5 | −1.475245 | 0.994979 | 1.204565 | 0.116051 | −0.50998 | −0.330371 |
| Eomes | −0.005112 | 0.362201 | 1.517082 | −1.559944 | 0.048375 | −0.362601 |
| Tcf7 | −0.562937 | 0.066607 | 1.970539 | −0.522208 | −0.289205 | −0.662796 |
| Ccr7 | −0.893954 | −0.163325 | 1.969748 | −0.395389 | −0.21246 | −0.30462 |
| Cd86 | 1.380865 | 1.183911 | −0.546108 | −0.52407 | −0.777066 | −0.717531 |
| Btla | 0.971706 | 0.874718 | 0.860138 | −0.76399 | −1.171717 | −0.770855 |
| Sik1 | 0.195164 | 0.195164 | 0.195164 | 1.116809 | 0.195164 | −1.897467 |
| Epcam | 1.063844 | 0.498488 | 0.93633 | −1.165034 | −1.165034 | −0.168594 |
| Egr1 | −0.013398 | −0.699301 | 1.896141 | −0.699301 | 0.151141 | −0.635282 |
| Ifit3 | 1.114624 | 0.9678 | 0.615644 | −0.896586 | −0.812924 | −0.988558 |

respiratory route can cause systemic infections that may pass through the lung draining LN. For example, replication-competent vaccinia virus has been detected in the mLN after respiratory infection (*Yates et al., 2008*). In such cases, the second line of Trm in the mLN may provide additional protection from the development of systemic disease after reinfection that manages to escape the lung-localized immune response. Additionally, mLN Trm have recently been shown to give rise to new populations of lung Trm after influenza rechallenge (*Paik and Farber, 2021*), thus serving as a reservoir capable of sensing infections that originate in the lung and providing amplified defenses back to the critical organ. Whether these 4M Trm cells in the mLN provide additional coordinated systemic responses or serve as a source for lung Trm upon additional infections cells as seen for 1M lung Trm (*Paik and Farber, 2021*) warrants further investigation.

Overall, our findings suggest that repeated lung infections or immunizations will preferentially enhance a durable population of Trm cells capable of protection against a viral insult in situ within the draining mLN.

# Materials and methods

## Key resources table

| Reagent type (species) or resource | Designation | Source or reference | Identifiers | Additional information |
|---|---|---|---|---|
| Strain, strain background (*Mus musculus*) | C57BL6/J | Jackson Laboratory | Stock No: 000664 (RRID:IMSR_JAX:000664) | |
| Strain, strain background (*Mus musculus*) | B6.PL(84NS)/Cy | Jackson Laboratory | Stock No: 000983 (RRID:IMSR_JAX:000406) | C57BL6/J Thy1.1 |
| Strain, strain background (*Mus musculus*) | B6.Cg-Tcra$^{tm1Mom}$ Tg(TcrLCMV)327Sdz (P14) | Jackson Laboratory | Stock No: 37394-JAX (RRID:IMSR_TAC:4138) | |
| Strain, strain background (*Mus musculus*) | Thy1.1/1.1-B6.Cg-Tcra$^{tm1Mom}$ Tg (TcrLCMV)327Sdz | This paper | Thy1.1/1.1 P14 | Can be acquired through lab contact or breeding of above commercially available strains |
| Strain, strain background (*Mus musculus*) | Thy1.1/1.2-B6.Cg-Tcra$^{tm1Mom}$ Tg (TcrLCMV)327Sdz | This paper | Thy1.1/1.2 P14 | Can be acquired through lab contact or breeding of above commercially available strains |
| Strain, strain background (Influenza A virus) | Recombinant influenza A/PR/8/34 expressing (H1N1) GP$_{33-41}$ | Laidlaw et al. Cooperativity Between CD8+ T Cells, Non- Neutralizing Antibodies, and Alveolar Macrophages Is Important for Heterosubtypic Influenza Virus Immunity. Plos Pathog. 9(3) e1003207 (2013). | PR8-GP33 | Can be acquired through lab contact. |
| Strain, strain background (Influenza A virus) | Recombinant influenza A/X-31 (H3N2) expressing GP$_{33-41}$ | Laidlaw et al. Cooperativity Between CD8+ T Cells, Non- Neutralizing Antibodies, and Alveolar Macrophages Is Important for Heterosubtypic Influenza Virus Immunity. Plos Pathog. 9(3) e1003207 (2013). | X31-GP33 | Can be acquired through lab contact. |
| Peptide, recombinant protein | GP$_{33-44}$ | AnaSpec | Catalog #: AS-61296 | |
| Antibody | CD11a (rat monoclonal) | Biolegend | M17/4 (AB_312776) | FACs (1:100) |
| Antibody | IFNγ (rat monoclonal) | eBioscience | XMG1.2 (AB_465410) | FACs (1:100) |
| Antibody | CD8a (rat monoclonal) | eBioscience | 53–6.7 (AB_1853141) | FACs (1:100) |
| Antibody | Thy1.1 (mouse monoclonal) | eBioscience | OX-7 (AB_2201314) | FACs (1:100) |
| Antibody | Thy1.2 (rat monoclonal) | eBioscience | 30-H12 (AB_1853152) | FACs (1:100) |
| Antibody | CD45.2 (mouse monoclonal) | eBioscience | 104 (AB_469724) | FACs (1:100) |
| Antibody | CD103 (hamster monoclonal) | Biolegend | 2E7 (AB_469040) | FACs (1:100) |
| Antibody | CD69 (hamster monoclonal) | Biolegend | H1.2F3 (AB_1853105) | FACs (1:100) |
| Antibody | CD44 (rat monoclonal) | Biolegend | 1M7 (AB_223593) | FACs (1:100) |
| Antibody | CD62L (rat monoclonal) | Biolegend | MEL-14 (AB_1853103) | FACs (1:100) |

*Continued on next page*

*Continued*

| Reagent type (species) or resource | Designation | Source or reference | Identifiers | Additional information |
|---|---|---|---|---|
| Antibody | KLRG1 (mouse monoclonal) | eBioscience | 2F1 (AB_540279) | FACs (1:100) |
| Antibody | CX3CR1 (mouse monoclonal) | eBioscience | SA011F11 (AB_2565701) | FACs (1:100) |
| Antibody | CXCR3 (Armenian hamster monoclonal) | eBioscience | CXCR3-173 (AB_1210593) | FACs (1:100) |
| Antibody | Eomesodermin (rat monoclonal) | eBioscience | Dan11mag (AB_11042577) | FACs (1:100) |
| Antibody | TNF (rat monoclonal) | eBioscience | MP6-XT22 (AB_465416) | FACs (1:100) |
| Antibody | IL-2 (rat monoclonal) | Biolegend | JES6-5H4 (AB_315298) | FACs (1:100) |
| Antibody | Granzyme A (mMouse monoclonal) | Biolegend | 3G8.5 (AB_2565308) | FACs (1:100) |
| Antibody | Granzyme B (rat monoclonal) | Biolegend | 12F9B65 (AB_2564373) | FACs (1:100) |
| Antibody | BrdU (mouse monoclonal) | Biolegend | Bu20a (AB_1595472) | FACs (1:100) |
| Commercial assay or kit | Foxp3/Transcription Factor Staining Buffer Set | Invitrogen | 00-5523-00 | |
| Software, algorithm | GraphPad Prism | GraphPad Prism 8 | Version 8.4.2 (464) (RRID:SCR_002798) | |

## Mice

C57BL/6 (Thy1.2/Thy1.2) mice were purchased from the National Cancer Institute (Frederick). P14 (Thy1.1/Thy1.1) TCR transgenic mice on a C57BL/6 background were originally acquired from Michael Bevan (University of Washington) and maintained in-house (*Pircher et al., 1989*). Thy1.1/Thy1.2 heterozygous P14 TCR-Tg mice were generated and maintained in-house. Mice used in the experiments were female 6–20 weeks of age. All animal studies and procedures were approved by the University of Iowa Animal Care and Use Committee, under U.S. Public Health Service assurance, Office of Laboratory Animal Welfare guidelines.

## Adoptive transfer of P14 and memory generation

For the generation of 1M, peripheral blood or splenocytes from naive P14 TCR-transgenic mice were isolated, washed, and characterized by flow cytometry for the frequency of Vβ8.1,8.2 + Vα2 + P14 TCR-Tg T cells. Donor P14 mice with P14 frequencies <20% of total blood or splenocytes were excluded from transfers. RBC-lysed (Vitalyse) cell populations containing $10^4$ naive Thy1.1 P14 cells were diluted within sterile saline and transferred (200 µL total volume) via iv injection into naive C57BL/6 (Thy1.2) recipients. For the transfer of memory P14 cells, influenza PR8-GP33 immune mice were euthanized 90 days post-infection, spleens were forced through a 70-µm cell strainer, RBC-lysed, made into a single-cell suspension and stained with anti-Thy1.1-PE (clone Ox-7, Biolegend, San Diego) in phosphate-buffered saline (PBS) with 5% FCS and PE-labeled P14s were eluted following positive enrichment with magnetic anti-PE beads (Miltenyi Biotec, San Diego; *Wirth et al., 2010*). Purity of the population after enrichment was assessed by flow cytometry and ranged from 70 to 85%. To generate higher order memory, a cell mixture containing $10^5$ memory Thy1.1 P14 cells was injected into naive Thy1.2 C57BL/6 mice. 2M, 3M, and 4M memory P14 responses were initiated by IN infection of recipient mice with the same dose of PR8-GP33 24 hr after the adoptive transfer. Splenic populations consisting of low frequencies of memory P14s (P14s <0.2% of splenic CD8$^+$ T cells) were excluded from enrichments and experiments. To generate mice containing mixed populations of 1M and 4M mice, congenically distinct $10^4$ naive and $10^5$ 3M

cells were isolated as described above and co-transferred to naive Thy1.2/Thy1.2 C57BL/6 mice 24 hr prior to subsequent PR8-GP33 infection.

## Infection and immunizations and influenza virus titers

Mice were immunized by IN infection with a sublethal dose of recombinant influenza PR8-GP33 virus ($2 \times 10^4$ TCID50)(*Mueller et al., 2010*), or a sublethal dose ($2 \times 10^4$ TCID50) of recombinant X31-GP33 (*Mueller et al., 2010*). Both viruses were grown on chicken eggs. Virus containing allantoic fluid was diluted in PBS, and mice were inoculated with the specific viral dose after induction of anesthesia by ketamine + xylazine (80–100 mg/kg + 10–120.5 mg/kg) injection.

## Tissue preparation, T cell analysis, and flow cytometry

For quantification of antigen-specific memory CD8$^+$ T cells, influenza immune mice at the indicated time points post-infection were administered 2–3 µg of fluorescently labeled anti-CD45.2 antibody (IV exclusion) and euthanized 3 min later. Spleen and PBL were harvested and processed into single-cell suspensions. Lungs were cut into small pieces, incubated for 1 hr at 37°C in the presence of type I collagenase (125 U/ml), and deoxyribonuclease (0.1 mg/ml). Digested lungs were pressed through a 70-µm cell strainer and lymphocytes were enriched by centrifugation in 35% Percoll (GE Healthcare) diluted in HBSS with subsequent RBC lysis with Vitalise (CMDG). For flow cytometry analyses, single-cell suspensions were incubated for 30 min at 4°C with the indicated antibodies, washed, fixed (BD fix buffer), washed, and resuspended in FACS buffer for analysis. Samples were stained for Thy1.1 to identify P14 cells or in house prepared MHC class I tetramers for Db-NP366-374 and analyzed by flow cytometry to identify transgenic P14 cells or endogenous influenza specific T cells and expression of the indicated markers detected with anti-CD8 (clone 53–6.7, eBioscience), anti-Thy1.2 (clone 30-H12, eBioscience), anti-Thy1.1 (clone OX-7, eBioscience), anti-CD45.2 (clone 104, BioLegend), anti-CD103(clone 2E7, BioLegend), anti-CD69 (clone H1.2F3, BioLegend), anti-CD44 (clone 1M7, BioLegend), anti-CD11a (clone M17/4, eBioscience), anti-CD62L (clone MEL-14, BioLegend), anti-KLRG1 (clone 2F1, eBioscience), anti-CX3CR1 (clone SA011F11), anti-CXCR3 (clone CXCR3-173, eBioscience), anti-CX3CR1 (clone SA011F11, BioLegend), and anti-Eomes (clone Dan11mag, eBioscience). Flow cytometry data were acquired in FACS Diva using an LSR Fortessa (Becton Dickinson) and analyzed using FlowJo software (FlowJo, LLC.).

## Parabiotic surgery

Parabionts were surgically conjoined as previously described (*Van Braeckel-Budimir et al., 2018*). 90 days after PR8-GP33 infections, mice bearing 1M (Thy1.1/1.1) and 4M (Thy1.1/1.2) P14 cells were surgically conjoined. Parabionts were cohoused for 2 weeks prior to surgery and hair was extensively removed under ketamine anesthesia from 1 cm above the elbow to 1 cm below the knee on opposing sides of pairs of mice 1–2 days prior to parabiotic surgery. On the day of surgery, mice were anesthetized using ketamine/xylazine and injected subcutaneously with Meloxicam to induce analgesia. The shaved skin was aseptically cleaned with betadine and alcohol, mice were placed on a heating pad, covered with a surgical drape, and placed on their sides, with adjacent shaved areas facing up. Longitudinal skin incisions were performed, skin was gently detached from the subcutaneous tissue and the separation was performed along both sides of the entire incision. Knee and elbow joints were attached with non-absorbable 3–0 sutures, the skin of the two animals was connected with a continuous absorbable 5–0 suture starting ventrally from the elbow toward the knee with a reciprocal dorsal continuous suture. Bupivacaine was applied locally and each mouse was subcutaneously injected with sterile saline to prevent dehydration. To prevent bacterial infections, animals were fed an antibiotic-containing diet starting from day 2 prior to surgery until day 10 post-surgery. Mouse recovery was followed daily for 2 weeks after the procedure.

## LCMV challenge and FTY720 treatment

Naïve, 1M, or 4M mice were challenged with a standard dose of LCMV-Armstrong ($2.0 \times 10^5$ PFU, 200 µL) via i.p. injection. Tissues were aseptically removed 72 hr post-infection and homogenized within limited volumes of RP10 without FCS (0.5 mL for mLN, 1.0 mL for Spleen) with sterile glass slides in small tissue culture plates. Tissue homogenate was snap frozen in liquid nitrogen, samples underwent a clarification spin and supernatants were analyzed for LCMV viral titers as previously

described (*Olson et al., 2012*). To inhibit cellular recruitment to lymphoid tissues, naive or recipient mice were treated with FTY720 (1 mg/kg) injected i.p. each day (*Shiow et al., 2006*; Sigma-Aldrich).

## ICS and BrdU staining

For intracellular cytokine staining, single-cell suspensions were generated as previously described, $2.0–3.0 \times 10^6$ cells (cells/mL) were aliquoted into round-bottomed 96-well plates and stimulated for 5 hr at 37 Deg C with cognate (GP33) or irrelevant (Ova) peptides (2 µm/mL) in the presence of Brefeldin A. Where indicated, antibodies against CD107a/b were added for the duration of the stimulation to denote cellular degranulation. Cells were stained for cell surface antigens, fixed, permeabilized (BD) and stained for IFNγ, TNF, IL-2, Granzyme A, and Granzyme B. For the detection of in vivo cellular division, BrdU (Sigma-Aldrich, 1 mg/mouse) was injected i.p. daily for 5 consecutive days. Detection of BrdU incorporation was preformed per manufacturers protocol (BD Biosciences) and stained with anti-BrdU as previously described (*Kurup et al., 2019*).

## Whole LN explant imaging

B6 mice containing a mixed population of 1M and 4M P14s were injected via IV injection with a bolus of antibodies (20 µg αThy1.1-PE, 20 µg αThy1.2-APC) in sterile saline. 5–8 hr post-injection, lung draining mLN and non-draining iLN were isolated, connective tissue was removed and these nodes were adhered to the bottom of a tissue culture plate with tissue adhesive and filled with sterile filtered media (HBSS no phenol red, 10% FCS). All images were acquired on an SP8 NLO Microscope (Leica) using a 25× motor collar-corrected water immersion objective (1.0 NA). High-resolution stacks (512 × 512 format) of 50–200 xy sections sampled with 5–7 µm spacing with bidirectional scanning were acquired at an acquisition rate of 120–240 s per stack and merged with Leica X software. All images were acquired sequentially using the following Excitation/Emission parameters: Thy1.1-PE 930/565–605, Secondary Harmonic Generation (SHG), 930/435–485 nm, and Thy1.2-APC 633/650–700. To reduce noise images were acquired with a line averaging of 6 and a post-acquisition kernel-3 median filter was utilized. Sequences of acquired image stacks were transformed into volume-rendered 3D images with Imaris version 9.1. The surface function was utilized to determine PE-Thy1.1+P14 s with consistent thresholds for intensity (80) and size (above 50 voxels). The co-localization feature of Imaris was utilized to build a new channel denoting co-staining of Thy1.1-PE and Thy1.2-APC with consistent thresholds (112 for PE, 116 for APC), and the surface function was used on this co-localization channel to determine the 4M P14s (PE-Thy1.1$^+$/APC-Thy1.2$^+$) with consistent thresholds for intensity (*Kumar et al., 2017*) and size (above 50 voxels).

## FACS for RNAseq

IV-labeled CD45.2-BV450 (3 µg/mouse) was administered as described above via IV injection 3 min prior to sacrifice. Pooled splenocytes (6–10 samples/sort) or mLNs (15–25 samples/sort) were isolated from 1M or 4M mice at days 23–30 post-PR8-GP33 infection, processed as above, stained with APC labeled -CD19, -NK1.1, and -CD4, and CD103-PE, CD69-PE-Cy7, CD90.1-PerCP-eFluor710, Live/Dead-e780. Samples were negatively enriched with APC-beads and Macs columns (Miltenyi Biotec), resuspended in MACS buffer + DNAse for FACS. Populations of IV$^-$ CD69$^-$/CD103$^-$ (Tcirc) and IV$^-$ CD69$^+$/CD103$^+$ (Trms) were sorted and RNA was isolated for bulk RNA-sequencing.

## RNAseq

Freshly sorted 1M or 4M P14s were (as indicated above) were resuspended in TRIzol and RNA was purified using an RNAeasy kit (QIAGEN) according to the manufacturer's protocol. RNA was assessed for purity and quality using an Agilent 2100 Bioanalyzer and RNA seq was performed using the single cell/low input library kit (NEB), full-length cDNAs were sequenced on the Illumina HiSeq 2500 High-output platform using 2 × 150 paired-end libraries. The sequence reads quality was checked using FastQC (http://www.bioinformatics.babraham.ac.uk/projects/fastqc/) and were aligned to the mouse genome version mm10 using STAR aligner (*Dobin et al., 2013*). Following read alignment, gene expression profiles were computed using featureCounts (*Liao et al., 2014*). Filtering and visualization of differentially expressed genes with a log2fold change >1.5 < −1.5 and a p value < 0.05 were identified using Partek GS software and for some figures values from Partek were normalized and displayed using GraphPad Prism. Gene set enrichment and functional

assignment were performed in DAVID bioinformatics resources and software from the Broad Institute as described (*Martin and Badovinac, 2016*; *Shan et al., 2017*; *Subramanian et al., 2005*). Tissue resident memory gene set (*Table 2*) was developed from review of existing literature (*Urban et al., 2020*).

## Quantification and statistical analysis

Statistical differences between the two study groups were evaluated using an unpaired, two-tailed t-test. Statistical differences between more than two study groups (single factor) were evaluated using one-way ANOVA with Tukey's multiple comparison post hoc test. Two-way ANOVA with Sidak's multiple comparison post hoc test was used to assess comparison between more than two groups based on more than one parameter (multiple factors). Statistical significance was assigned as $*p<0.05$, $**p<0.01$, $***p<0.001$, and $****p<0.0001$. Statistical analyses were performed using Prism seven software (GraphPad).

## Acknowledgements

Funding: This work was supported by grants from the National Institutes of Health (AI42767 to JTH, AI114543 to JTH and VPB, A125446 and A127481 to NSB, AI124093 to SMV, GM134880 to VPB, AI121080 and AI139874 to H-HX, T32 5T32HL007 to SMA, T32 AI007511 to NVB-B and IJJ) and the Veteran Affairs BLR and D Merit Review Program (BX002903) to H-HX. The authors thank Lecia Epping and Lisa Hancox for technical assistance. The authors thank J Fishbaugh, H Vignes and M Shey (University of Iowa Flow Cytometry Core Facility) for cell sorting, I Antoshechkin (California Institute of Technology) and Admera health for RNA-seq. Data herein were obtained from the Flow Cytometry Facility, which is a Carver College of Medicine Core Research Facilities/Holden Comprehensive Cancer Center Core Laboratory at the University of Iowa.

## Additional information

### Funding

| Funder | Grant reference number | Author |
|--------|------------------------|--------|
| NIH Office of the Director | AI42767 | John Harty |
| NIH Office of the Director | AI11453 | Vladimir P Badovinac John Harty |
| NIH Office of the Director | AI25446 | Noah Butler |
| NIH Office of the Director | AI27481 | Noah Butler |
| NIH Office of the Director | AI124093 | Steven M Varga |
| NIH Office of the Director | GM134880 | Vladimir P Badovinac |
| NIH Office of the Director | AI121080 | Hai-Hui Xue |
| NIH Office of the Director | AI129874 | Hai-Hui Xue |
| NIH Office of the Director | T32HL007 | Scott M Anthony |
| NIH Office of the Director | T32AI1005511 | Natalija Van Braeckel-Budimir Isaac J jensen |
| Veterans Affairs Council, R.O.C. | BX002903 | Hai-Hui Xue |

The funders had no role in study design, data collection and interpretation, or the decision to submit the work for publication.

### Author contributions

Scott M Anthony, Conceptualization, Formal analysis, Investigation, Methodology, Writing - original draft, Writing - review and editing; Natalija Van Braeckel-Budimir, Conceptualization, Formal analysis, Investigation, Methodology, Writing - review and editing; Steven J Moioffer, Isaac J Jensen,

Formal analysis, Investigation, Writing - review and editing; Stephanie van de Wall, Stacey M Hartwig, Formal analysis, Investigation; Qiang Shan, Resources, Data curation, Formal analysis, Investigation; Rahul Vijay, Ramakrishna Sompallae, Software, Formal analysis; Steven M Varga, Formal analysis, Supervision, Funding acquisition; Noah S Butler, Software, Formal analysis, Supervision, Funding acquisition; Hai-Hui Xue, Resources, Supervision, Funding acquisition; Vladimir P Badovinac, John T Harty, Conceptualization, Resources, Formal analysis, Supervision, Funding acquisition, Writing - original draft, Writing - review and editing

## Author ORCIDs
Scott M Anthony http://orcid.org/0000-0002-1590-7401
Isaac J Jensen https://orcid.org/0000-0002-3107-3961
Noah S Butler https://orcid.org/0000-0002-1429-0796
Hai-Hui Xue http://orcid.org/0000-0002-9163-7669
Vladimir P Badovinac https://orcid.org/0000-0003-3180-2439
John T Harty https://orcid.org/0000-0001-7266-2802

## Ethics
Animal experimentation: All experimental procedures using mice were approved by the University of Iowa Animal Care and Use Committee under the ACURF protocol number (0051102). The experiments performed in this study were done under strict accordance to the Office of Laboratory Animal Welfare guidelines and the PHS Policy on Humane Care and Use of Laboratory Animals.

## Decision letter and Author response
Decision letter https://doi.org/10.7554/eLife.68662.sa1
Author response https://doi.org/10.7554/eLife.68662.sa2

# Additional files
## Supplementary files
• Transparent reporting form

## Data availability
RNA sequencing data has been deposited in GEO under accession code GSE172279. All other data generated and analyzed during this study are included in the source data files.

The following dataset was generated:

| Author(s) | Year | Dataset title | Dataset URL | Database and Identifier |
|---|---|---|---|---|
| Anthony SM, Moioffer SJ, van de Wall S, Shan Q, Sompallae R, Van Braeckel-Budimir N, Butler N, Hartwig SM, Xue HH, Harty J, Badovinac VP, Vijay R, Jensen IJ | 2021 | Bulk RNAseq of Sort-purified IV- Primary (1M) or Quaternary (4M) Tcirc (IV- CD69-/CD103-) or Trm (IV- CD69+/CD103+) Memory P14 cells isolated from Spleen or Mediastinal Lymph Nodes | https://www.ncbi.nlm.nih.gov/geo/query/acc.cgi?acc=GSE172279 | NCBI Gene Expression Omnibus, GSE172279 |

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
