## [Decision Letter]

**Acceptance summary:**

This is a very interesting study that undertakes a systematic evaluation of the cellular protective abilities of primary and quaternary stimulated CD8^+^ T cells. It reveals that repeated activation of virus-specific CD8 T cells can afford enhanced protection compared with cells generated following a single infection. The authors provide a detailed molecular analyses of these different cell types of responses. The implication of this study is that successive infections, such as occurs yearly from seasonal influenza, can enhance the immune response and protection.

**Decision letter after peer review:**

Thank you for submitting your article "Protective function and durability of lymph node resident memory CD8^+^ T-cells." for consideration by *eLife*. Your article has been reviewed by 3 peer reviewers, one of whom is a member of our Board of Reviewing Editors, and the evaluation has been overseen by Satyajit Rath as the Senior Editor. The reviewers have opted to remain anonymous.

Essential revisions:

1) Please address all the major points raised by the reviewers. This mainly expands explanations and discussions of findings.

2) Please consider the points raised by Reviewer 2 which may require provision of additional data.

*Reviewer #1:*

Cross-reactive CD8^+^ T cells represent a possible avenue for protection against heterologous influenza infections, yet the role and mechanisms of CD8-mediated protection during influenza infection are incompletely understood. In this manuscript, Anthony et al. show that repeated influenza exposure generates memory CD8^+^ T cells that better protect the lung draining mediastinal lymph node than cells that haven't been repeatedly exposed to antigen. To do this, the authors used a previously described system in which they adoptively transfer memory CD8^+^ T cells into naive hosts and infect those hosts with influenza, generating memory CD8^+^ T cells with 4 exposures (termed 4M cells) to influenza. Repeated Ag exposure generates lymph node-resident memory T cells that have different characteristics than those generated by a single exposure, including altered cytokine production and distribution. Further, repeated exposure results in a population of splenic memory CD8^+^ T cells genetically poised to become TRM. Thus, repeated antigen exposure results in enhanced generation of LN-protective TRM.

In general, this is a well written and executed study with substantiated conclusions. The authors use a complicated adoptive transfer system to experimentally address the repeated virus exposures encountered by humans that are difficult to otherwise model in mice. The data showing changes in gene expression after repeated exposure reflecting a TRM-core signature are exciting and raise new questions regarding the maintenance of memory in the tissue.

There are a few revisions that would enhance the message of the paper.

The authors demonstrate that repeated antigen exposure results in a number of alterations in T cells present in the lymph node (LN), including the number of cells present and their phenotype, the production of granzyme, and the distribution of cells within the LN. Although the latter two are difficult to examine experimentally, the authors could determine the role of total T cell number in LN protection. Is the killing capacity the same on a per cell basis?

The manuscript would benefit from enhanced analyses of the differences in T cell location in the LN between 1M and 4M cells (Figure 5). The precise areas of the LN in which the different cells reside is important to know. Are the cells differentially distributed in the T cell zone or the subcapsular sinus, etc.? This has implications for not only effector function of the cells but also trafficking from the LN.

An important conclusion of this study is that TRM in the lung-draining LN are important for the control of nodal viral infection, which the authors demonstrate using an LCMV challenge. However, many viruses, including influenza, do not replicate in the draining LN (as the authors note). This should be mentioned in the discussion as it is unlikely that, at least for influenza, nodal protection is the reason for the presence of TRM in the draining LN.

Figure 2. D-E. The authors suggest that control of LCMV in the LN may decrease virus spread to the spleen. Do the authors believe that LCMV is disseminating lymphohematogenously from the LN to the spleen? If so, please demonstrate this. It seems more likely that 4M T cells are better controlling virus in the spleen as well as the LN.

Please include a discussion of why it would be important to control infection in the mediastinal LN? Some examples here would be appropriate. What viruses can replicate there and how does this contribute to pathogenesis (as the virus has already been exposed to the respiratory epithelium)?

As discussed above, one of the main points of the paper is the increased protective capacity of 4M T cells in the LN, however the mechanism is unclear and could be: increased cytolytic function, altered distribution, or increased numbers of LN 4M cells (as shown in Figure 2). If numbers of T cells were adjusted such that the number of 1M cells in the tissue at the time of challenge was equal to the number of 4M, would protection (viral titers) be the same in the LN? Is the capacity for protection different on a per cell basis or is the advantage seen with 4M due to the increase in cell numbers? This could be done either through transfer of differential numbers of cells or waiting later for there to be a similar number of cells, as shown in Figure 3.

*Reviewer #2:*

Anthony et al., investigate the development and biological relevance of resident memory T cells that deposit in the lung draining lymph node (LN) following respiratory infection. They show that repeated antigen exposure generates a circulating memory T cell pool enriched for cells which have an enhanced capacity to develop into LN resident memory T cells. Moreover, this group shows that these LN Trm can help facilitate the clearance of a local viral infection. The biological relevance of these findings (ie the antiviral control of LN resident Trm) in the context of IAV control is lacking as influenza is generally not thought to replicate actively in lymph nodes. Nonetheless, this report is interesting, expands on the earlier findings reported by this group on lung Trm (Van Braeckel-Budimir, Cell Reports, 2018) and provides important insight into how multiple antigen encounters can change the phenotype, functionality, longevity and fate of the memory T cell pool. This knowledge is important and has implications for the development of T cell-based vaccines. The data presented in this paper is robust and supports the conclusions made by the authors.

1. Figure 1 c- To support their claim that intranasal infection is needed for mLN Trm formation I recommend the authors perform an i.p. injection with influenza virus (PR8-gp33) and check mLN Trm development. This would represent a more suitable control to the data presented in Figure 1c.

2. Figure 2 – Can the authors show the activation status, cytokine profile, proliferation (ie BrdU) and numbers of 1M and 4M mLN Trm pre and post LCMV challenge as this will provide insight into how infection control is mediated.

3. Figure 5 – it would be interesting to compare localisation of the P14 1M v 4M in the LN shown in Figure 5 to localisation following LCMV infection (potentially relative to LCMV infected cells).

4. Can the authors comment on how they think multiple antigen exposures is refining the memory pool and enriching for these Trm precursors (ie Is it selective expansion of these precursor cells and is this reflected in a skewing of the clonality of the memory pool?).

*Reviewer #3:*

This study aims to determine if successive infections enhance protective lung infection and to delineate the transcriptional profile of these cells via RNAseq analyses.

The authors perform a very systematic study. They provide an important explanation of the model and its limitations in evaluating the protective capacity of CD8^+^ T cells. By modifying these previous models they are able to provide insight to the cellular changes that occur through successive activation. By examining these cells molecularly, the find the not so surprising finding that the tissue-resident cells did not exhibit striking differences between the primary stimulated and quaternary stimulated cells. On examination of the circulating CD8^+^ T cells, however, they observed that they were enriched with transcripts normally expressed in tissue-resident cells. Intriguingly, a core set tissue-resident associated genes was upregulated in the circulating cells suggesting that repeated antigen activation generates circulating cells that are poised to adopt a tissue-resident lifestyle.

In general, this is a very thorough study. The authors perform both cellular and molecular analyses, establishing a blueprint for the 'poised' phenotype of the 4M cells. Unfortunately, the authors do not define or test any candidate molecules, or provide any discussion around this imprint. It would be very interesting to test a number of these candidates. Outside the scope of the work, this leads to the question of the blueprint of epigenetic changes that are induced through the multiple stimulations.

This could be expanded in the manuscript with discussion of the potential models of regulation of these cells.

---

## [Author Response]

Essential revisions:1) Please address all the major points raised by the reviewers. This mainly expands explanations and discussions of findings.

We have addressed each point below.

2) Please consider the points raised by Reviewer 2 which may require provision of additional data.

We have considered the points raised by R2 and have modified the manuscript in response to limit conclusions, provide clarification, and add discussion. The reviewer raises some interesting points about relative killing capacity, but given that the cells of interest are Trm in the mLN, we are at a loss as to how to derive the necessary E:T information prior to challenge and we cannot normalize numbers by adoptive transfer. In our hands, sorting cells and asking them to kill in vitro is a poor experiment, based on the damage accrued during sorting that takes time for cellular recovery.

Reviewer #1:[…] There are a few revisions that would enhance the message of the paper.The authors demonstrate that repeated antigen exposure results in a number of alterations in T cells present in the lymph node (LN), including the number of cells present and their phenotype, the production of granzyme, and the distribution of cells within the LN. Although the latter two are difficult to examine experimentally, the authors could determine the role of total T cell number in LN protection. Is the killing capacity the same on a per cell basis?

Please see response our last response to reviewer #1 below with new text pp8, lines 193-195.

The manuscript would benefit from enhanced analyses of the differences in T cell location in the LN between 1M and 4M cells (Figure 5). The precise areas of the LN in which the different cells reside is important to know. Are the cells differentially distributed in the T cell zone or the subcapsular sinus, etc.? This has implications for not only effector function of the cells but also trafficking from the LN.

This is a good point. We highlight this point with a new citation and clarify that more work needs be done to address this issue, pp 9, lines 206-209.

An important conclusion of this study is that TRM in the lung-draining LN are important for the control of nodal viral infection, which the authors demonstrate using an LCMV challenge. However, many viruses, including influenza, do not replicate in the draining LN (as the authors note). This should be mentioned in the discussion as it is unlikely that, at least for influenza, nodal protection is the reason for the presence of TRM in the draining LN.

We have added additional discussion to the manuscript to address this point, pp15, lines 360-369. Please also see response to our fifth response to reviewer #2 below.

Figure 2. D-E. The authors suggest that control of LCMV in the LN may decrease virus spread to the spleen. Do the authors believe that LCMV is disseminating lymphohematogenously from the LN to the spleen? If so, please demonstrate this. It seems more likely that 4M T cells are better controlling virus in the spleen as well as the LN.

We qualify our conclusions to more accurately discuss the data on this point in the revised manuscript, pp7, lines 145-151.

Please include a discussion of why it would be important to control infection in the mediastinal LN? Some examples here would be appropriate. What viruses can replicate there and how does this contribute to pathogenesis (as the virus has already been exposed to the respiratory epithelium)?

We have added additional discussion to the manuscript to address this point, pp15, lines 360-369.

As discussed above, one of the main points of the paper is the increased protective capacity of 4M T cells in the LN, however the mechanism is unclear and could be: increased cytolytic function, altered distribution, or increased numbers of LN 4M cells (as shown in Figure 2). If numbers of T cells were adjusted such that the number of 1M cells in the tissue at the time of challenge was equal to the number of 4M, would protection (viral titers) be the same in the LN? Is the capacity for protection different on a per cell basis or is the advantage seen with 4M due to the increase in cell numbers? This could be done either through transfer of differential numbers of cells or waiting later for there to be a similar number of cells, as shown in Figure 3.

We thank the reviewer for raising this point. While we do believe that numbers of Trm in the mLN matter, we cannot rule out functional differences between 1M and 4M Trm. Unfortunately, given that the cells in question are Trm and reside in tissues, we cannot perform simple adoptive transfer studies to normalize their numbers. Additionally, we cannot normalize their numbers by selecting specific mice (although we tried based on circulating frequencies, see Figure 2A) prior to challenge infection because we can only access the blood at this timepoint and as shown, equivalent blood frequencies do not reflect equivalent Trm in the mLN (Figure 2A-C). Thus, it would likely take many, many mice to find the few with similar numbers of 1M vs 4M Trm in the mLN, which we could only assess once, either before or after challenge. This limitation is now addressed in the revised manuscript, pp 8, lines 193-194

Reviewer #2:[…] 1. Figure 1 c- To support their claim that intranasal infection is needed for mLN Trm formation I recommend the authors perform an i.p. injection with influenza virus (PR8-gp33) and check mLN Trm development. This would represent a more suitable control to the data presented in Figure 1c.

We thank the reviewer for this point. However, unless we do all peripheral infections and immunizations (such as flu i.p. which does not replicate) we can never rule out that a specific infection will not generate mLN Trm. We modify our conclusion to state the limitations and clarify that point in the revised manuscript, pp 5, lines 113-119.

As an aside, it is a bit puzzling that we see no CD69+CD103+ Trm in the mLN, since LCMV does replicate in the mLN, but that investigation is beyond the scope of the current work.

2. Figure 2 – Can the authors show the activation status, cytokine profile, proliferation (ie BrdU) and numbers of 1M and 4M mLN Trm pre and post LCMV challenge as this will provide insight into how infection control is mediated.

We thank the reviewer for this important point and have modified the manuscript to clarify that these are important experiments to do in the future, pp 7, lines 145-151.

3. Figure 5 – it would be interesting to compare localisation of the P14 1M v 4M in the LN shown in Figure 5 to localisation following LCMV infection (potentially relative to LCMV infected cells).

This is a good point also raised by R#1 about how precise localization in the mLN could influence protection. We highlight this point with a new citation and clarify that more work needs be done to address this issue, pp 9, lines 206-209.

4. Can the authors comment on how they think multiple antigen exposures is refining the memory pool and enriching for these Trm precursors (ie Is it selective expansion of these precursor cells and is this reflected in a skewing of the clonality of the memory pool?)

We have added discussion, pp 13, lines 314-320.

Reviewer #3:[…] In general, this is a very thorough study. The authors perform both cellular and molecular analyses, establishing a blueprint for the 'poised' phenotype of the 4M cells. Unfortunately, the authors do not define or test any candidate molecules, or provide any discussion around this imprint. It would be very interesting to test a number of these candidates.

We agree, although we respectfully suggest those studies are also outside of the scope of the current manuscript.

Outside the scope of the work, this leads to the question of the blueprint of epigenetic changes that are induced through the multiple stimulations.This could be expanded in the manuscript with discussion of the potential models of regulation of these cells.

We have added discussion to address the reviewer’s general point, pp13 lines 314-320.